# Pines' demon observed as a 3D acoustic plasmon in Sr$_2$RuO$_4$

Ali A. Husain[1✉], Edwin W. Huang[2], Matteo Mitrano[3], Melinda S. Rak[1], Samantha I. Rubeck[1], Xuefei Guo[1], Hongbin Yang[4], Chanchal Sow[5,9], Yoshiteru Maeno[5,6], Bruno Uchoa[7], Tai C. Chiang[1], Philip E. Batson[8], Philip W. Phillips[2] & Peter Abbamonte[1✉]

The characteristic excitation of a metal is its plasmon, which is a quantized collective oscillation of its electron density. In 1956, David Pines predicted that a distinct type of plasmon, dubbed a 'demon', could exist in three-dimensional (3D) metals containing more than one species of charge carrier[1]. Consisting of out-of-phase movement of electrons in different bands, demons are acoustic, electrically neutral and do not couple to light, so have never been detected in an equilibrium, 3D metal. Nevertheless, demons are believed to be critical for diverse phenomena including phase transitions in mixed-valence semimetals[2], optical properties of metal nanoparticles[3], soundarons in Weyl semimetals[4] and high-temperature superconductivity in, for example, metal hydrides[3,5–7]. Here, we present evidence for a demon in Sr$_2$RuO$_4$ from momentum-resolved electron energy-loss spectroscopy. Formed of electrons in the $\beta$ and $\gamma$ bands, the demon is gapless with critical momentum $q_c = 0.08$ reciprocal lattice units and room-temperature velocity $v = (1.065 \pm 0.12) \times 10^5$ m s$^{-1}$ that undergoes a 31% renormalization upon cooling to 30 K because of coupling to the particle–hole continuum. The momentum dependence of the intensity of the demon confirms its neutral character. Our study confirms a 67-year old prediction and indicates that demons may be a pervasive feature of multiband metals.

Proposed in 1952 by Pines and Bohm[8], plasmons were first observed in inelastic electron scattering experiments[9] and were one of the first confirmed examples of collective phenomena in solids. Landau referred to plasmons as 'zero sound', stressing that they are the quantum analogue of acoustic sound in a classical gas[10]. However, unlike ordinary sound, whose frequency tends toward zero at zero momentum, $q$ (that is, as its wavelength approaches infinity), plasmons, except in lower-dimensional systems, cost a finite energy to excite, as creating a density oscillation requires overcoming the long-ranged Coulomb interaction[1,8]. The plasma frequency, $\omega_p$, in ordinary metals ranges from 15 eV in Al (ref. 11) to 20 eV in Cu (ref. 12).

In 1956, Pines predicted that it was possible to create a plasmon excitation with no Coulomb energy cost[1]. The new collective mode, dubbed a 'demon', arises when electrons in different bands move out of phase, thereby resulting in no net transfer of charge but a modulation in the band occupancy. A demon may be thought of as a collective mode of neutral quasiparticles whose charge has been fully screened by electrons in a separate band. Applying the random phase approximation (RPA), Pines argued that the frequency of a demon mode, $\omega$, should scale as $\omega \approx q$, vanishing as $q \to 0$ (ref. 1).

Surprisingly, although discussed widely in the theoretical literature[1,2,5,6,13–15], there appears to be no experimental confirmation of a demon in a 3D metal, even 67 years after its prediction. Acoustic plasmons have been widely studied in two-dimensional (2D) metals[16–19], in which conventional, single-component plasmons are gapless[20]. Low-energy plasmons have also been reported in layered 3D metals at $q = \pi/d$ ($d$ being the layer spacing), mostly recently by resonant inelastic X-ray scattering techniques[21,22], although these excitations disperse to $\omega_p$ at $q = 0$ so are not acoustic[23]. A demon was once reported in photoexcited GaAs, though the effect was only transient[24]. A true demon, that consists of out-of-phase movement of distinct electron fluids and remains acoustic as $q \to 0$ in a 3D system, has not yet been reported.

If demons were shown to exist experimentally, a proper, many-body theory of demons, that incorporates hydrodynamics and beyond-RPA effects, would surely be needed.

What makes demons difficult to detect is their inherent charge neutrality. The out-of-phase currents of the two electron fluids exactly cancel as $q \to 0$, extinguishing the long-ranged part of the Coulomb interaction. For this reason, a demon has no signature in the dielectric function of a metal, $\varepsilon(q, \omega)$, in the limit of small $q$, and does not couple to light. The most promising way to detect a demon is to measure the excitations of a multiband metal at non-zero $q$, where a demon modulates the density and may be experimentally observable using electron energy-loss spectroscopy (EELS) techniques that observed plasmons originally[9].

[1]Department of Physics and Materials Research Laboratory, University of Illinois, Urbana, IL, USA. [2]Department of Physics and Institute for Condensed Matter Theory, University of Illinois, Urbana, IL, USA. [3]Department of Physics, Harvard University, Cambridge, MA, USA. [4]Department of Chemistry and Chemical Biology, Rutgers University, Piscataway, NJ, USA. [5]Department of Physics, Kyoto University, Kyoto, Japan. [6]Toyota Riken - Kyoto Univ. Research Center (TRiKUC), KUIAS, Kyoto University, Kyoto, Japan. [7]Department of Physics and Astronomy, University of Oklahoma, Norman, OK, USA. [8]Department of Physics, Rutgers University, Piscataway, NJ, USA. [9]Present address: Department of Physics, Indian Institute of Technology, Kanpur, India. ✉e-mail: husain.ali.abdullah@gmail.com; abbamont@illinois.edu

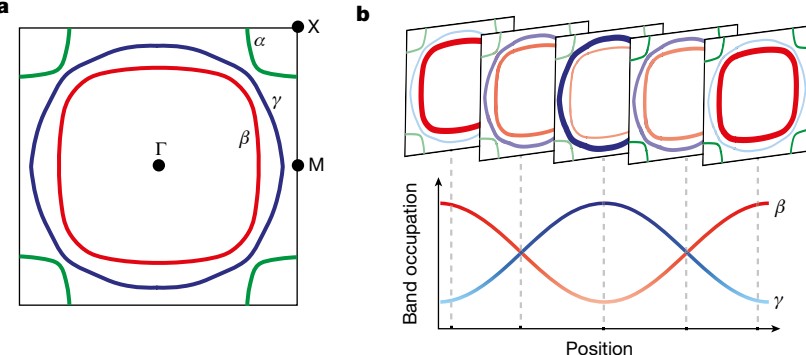

**Fig. 1 | Conceptual illustration of the demon excitation in Sr₂RuO₄. a**, Fermi surface showing the three species of electrons, $\alpha$, $\beta$ and $\gamma$. **b**, Conceptual illustration of the demon in $Sr_2RuO_4$, which is a modulation in the $\gamma$ and $\beta$ band fillings that keeps the overall electron density constant.

The metal we investigate is $Sr_2RuO_4$, which has three nested bands, $\alpha$, $\beta$ and $\gamma$, crossing the Fermi energy (Fig. 1a)[25,26]. At a temperature $T \lesssim 40$ K, $Sr_2RuO_4$ is a good Fermi liquid showing resistivity $\rho \approx T^2$, well-defined quantum oscillations[27] and the expected scattering rate in optics[28]. At higher temperatures, $T \gtrsim 600$ K, $Sr_2RuO_4$ crosses over into a strongly interacting 'strange metal' phase in which the quasi-particles are highly damped[29], the resistivity $\rho \approx T$ and its value exceeds the Mott–Ioffe–Regel limit at high temperature[30]. The strong interactions arise from Hund's coupling and are described well by dynamical mean field theory[26,31].

As a multiband metal, $Sr_2RuO_4$ is a candidate for exhibiting a demon. In particular, the $\beta$ and $\gamma$ bands have quite different velocities and curvature[25,26,32], reminiscent of Pines' original conceptualization of a demon as a mode in which light electrons screen the Coulomb interaction between heavy electrons[1]. Understanding whether a demon is expected in $Sr_2RuO_4$ requires a microscopic calculation.

We calculated the collective charge excitations of $Sr_2RuO_4$ by computing its dynamic charge susceptibility, $\chi(q, \omega)$, in the RPA[8,9,12] (see section 'Multiband RPA calculations' in Methods). RPA is an approximate theory for computing the collective modes of Fermi liquids that, although inexact, can yield insight into the number of excitations and their approximate energies. We first computed the Lindhard function using a tight-binding parameterization of the energy bands, and then determined the susceptibility, $\chi(q, \omega)$, using the Coulomb interaction $V(q) = e^2/\varepsilon_\infty q^2$, where $e$ is the electron charge and $\varepsilon_\infty = 2.3$ is the background dielectric constant taken from ref. 28. The calculation has no adjustable parameters and no fine tuning or fitting to experimental data was done.

Figure 2 shows the imaginary part, $\chi''(q, \omega)$ along the (1,0,0) direction as a function of momentum, $q$, and energy, $\omega$. The most prominent feature is a sharp plasmon at $\omega_p = 1.6$ eV (Fig. 2a), which is similar to the measured zero crossing of the real part of $\varepsilon(0, \omega)$ in optics[28]. The plasmon exhibits a downward dispersion, which is a band structure effect similar to that observed in transition metal dichalcogenides[33]. Note that the intensity of the plasmon (colour scale) scales as $q^2$ at small momenta (Fig. 2a), which is consistent with the $f$-sum rule[12]. This permits $\varepsilon(q, 0) = 1/[1 + V(q)\chi(q, 0)]$ to diverge at small values of $q$, which is required in a metal in which the electric field should be completely screened over long distances.

At low energy, the calculation also shows an acoustic mode (Fig. 2b). Its velocity, $v = 0.639$ eV Å, lies between the velocities of the $\beta$ and $\gamma$ bands, which is an expected property of a demon[1]. Unlike the plasmon, the intensity of this excitation scales as $q^4$ (Fig. 2b and Extended Data Fig. 10), which is faster than would be expected from the $f$-sum rule. Were this the only excitation present in the material, it would imply that $\varepsilon(q, 0) = 1/[1 + V(q)\chi(q, 0)] \to 1$ in the limit of small $q$, meaning that this excitation is neutral and does not contribute to screening over large distances.

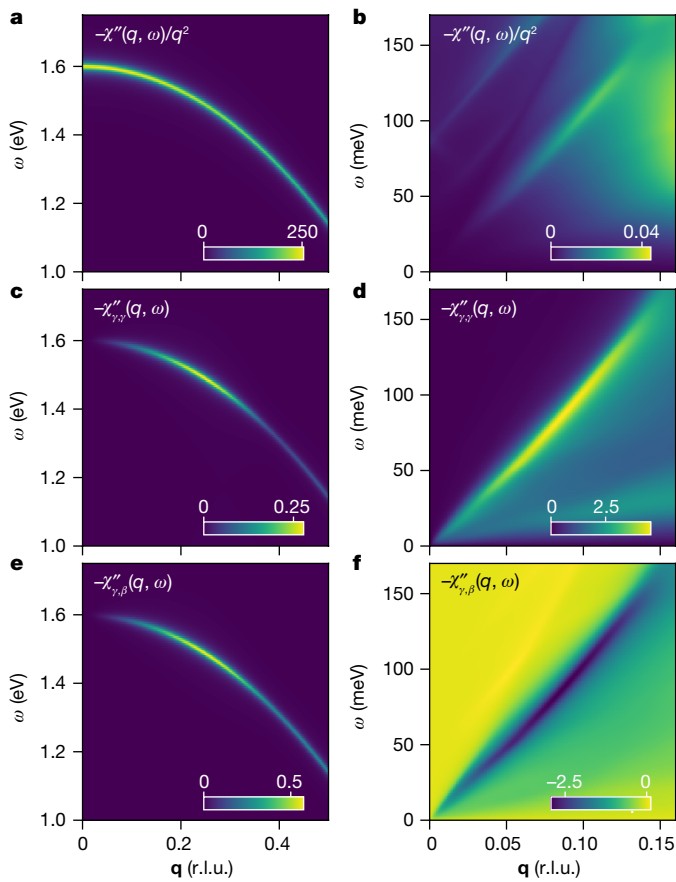

**Fig. 2 | Charge susceptibility of Sr₂RuO₄ from RPA. a**, Colour plot of the scaled charge susceptibility, $\chi''(q, \omega)/q^2$, for **q** in the (1,0,0) direction, showing that the intensity of the conventional, high-energy plasmon scales as $q^2$ as $q \to 0$. **b**, The same plot in the low-energy region, showing that the intensity of the demon goes to zero faster than $q^2$ in the same limit. **c**, Colour plot of the band-decomposed susceptibility, $\chi''_{s,s'}(q, \omega)$ (see Methods) for band indices $s = s' = \gamma$ in the vicinity of the plasmon. **d**, Same quantity as panel **a** in the region of the demon. **e**, Band-decomposed susceptibility for $s = \gamma$, $s' = \beta$ in the region of the plasmon. **f**, Same quantity as panel **e** in the region of the demon. The sign of the response demonstrates that $\gamma$ and $\beta$ electrons oscillate in phase for the conventional plasmon and out of phase for the demon.

This excitation is definitively identified as a demon by examining the partial susceptibilities, $\chi_{a,b}$, which describe the linear response of the density of electrons in band $a$ due to an external potential that couples only to electrons in band $b$. As explained in the section 'Band

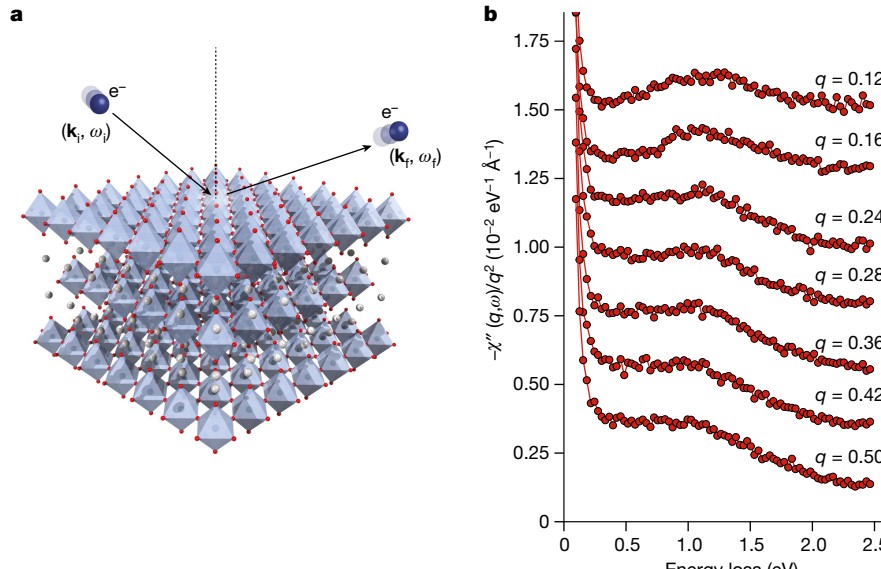

**Fig. 3 | High-energy M-EELS spectra from Sr₂RuO₄. a**, Conceptual illustration of reflection M-EELS experiments from a cleaved Sr₂RuO₄ surface. **b**, Fixed-*q* (in r.l.u.) energy-loss scans for a selection of *q* values along the (1,0) crystallographic direction, taken at *T* = 300 K. These spectra were obtained by

dividing the M-EELS matrix elements and scaling the curves as described in ref. 37. At small momenta (*q* < 0.16 r.l.u.), the spectra show a broad plasmon feature that peaks at 1.2 eV. At larger momenta, the data show an energy-independent continuum as was observed previously in Bi₂Sr₂CaCu₂O₈₊ₓ (ref. 37).

decomposition of the susceptibility' in Methods, the relative sign of $\chi''_{a,b}$ and $\chi''_{a,a}$ indicates whether electrons in the bands *a* and *b* oscillate in- or out of phase. For example, if we consider the plasmon (Fig. 2c and Extended Data Fig. 10b), the quantities $\chi''_{\gamma,\gamma}$, $\chi''_{\beta,\beta}$ and $\chi''_{\gamma,\beta}$ are all negative, meaning the *β* and *γ* subbands oscillate in phase, regardless of which is excited. The situation is different for the acoustic mode. Whereas $\chi''_{\gamma,\gamma}$ and $\chi''_{\beta,\beta}$ are both negative (Fig. 2d and Extended Data Fig. 10c), the off-diagonal term $\chi''_{\gamma,\beta}$ is positive (Fig. 2f), meaning that if one drives the *γ* electrons, the *β* electrons respond 180° out of phase. This demonstrates that the acoustic mode predicted in RPA is a true demon in that it consists of an out-of-phase oscillation between the *β* and *γ* electrons (Fig. 1b).

We now compare the RPA results to momentum-resolved electron energy-loss spectroscopy (M-EELS)[34] measurements of the collective excitations of Sr₂RuO₄ with an energy resolution Δ*ω* = 6 meV and momentum resolution Δ*q* = 0.03 Å⁻¹. M-EELS is done in reflection mode and measures both surface and bulk excitations at non-zero momentum transfer, *q* (ref. 34), where the signature of a demon should be clearest (Fig. 2b). Sr₂RuO₄ crystals were grown as described previously[35] and cleaved in situ in ultra-high vacuum to reveal pristine surfaces. The surfaces were passivated by exposing to residual CO gas, which disorders the √2*a* × √2*a* surface reconstruction[26] and terminates surface dangling bonds[26,36]. This treatment eliminates the surface state that complicated interpretation of early angle-resolved photoemission (ARPES) experiments[25,32] and results in bulk-like properties in surface measurements[26].

M-EELS spectra at *T* = 300 K at large energy transfer show a broad plasmon peak at approximately 1.2 eV (Fig. 3b, top curve). Its width at *q* = 0.12 reciprocal lattice units (r.l.u.) is approximately 10² larger than the predicted width of the 1.6 eV plasmon in RPA. This discrepancy is unsurprising as Sr₂RuO₄ is a non-Fermi liquid at *ω* ≳ 50 meV (refs. 26,28–31) and RPA neglects many interaction effects that could shift and damp the plasmon. Nevertheless, RPA correctly predicts its existence and approximate energy. At larger momenta, *q* ≳ 0.28 r.l.u, the plasmon evolves into a featureless, energy-independent continuum similar to that observed in Bi₂Sr₂CaCu₂O₈₊ₓ (Bi-2212)[37,38], although the cutoff energy in Sr₂RuO₄ is higher (1.2 eV compared with 1.0 eV in Bi-2212). This observation was confirmed by bulk, transmission EELS measurements

using a Nion UltraSTEM (Methods), establishing it as a bulk effect, and indicates that this continuum may be a generic feature of the *q* ≠ 0 density response of strange metals.

In the low-energy, Fermi liquid regime, M-EELS reveals an acoustic mode (Fig. 4). Its energy gap at *q* = 0 is less than 8 meV, an upper bound set by the tails of the elastic line (Methods). The dispersion of the mode in the (1,0) direction is linear over most of its range, with room-temperature group velocity *v*g = 0.701 ± 0.082 eV Å (which equals (1.065 ± 0.12) × 10⁵ m s⁻¹). At small momentum, *q* < 0.03 r.l.u., the dispersion shows a quadratic 'foot', in which *ω*(*q*) ≈ *q*², which is a real effect not caused by the finite *q* resolution of the measurement. The linewidth of the mode increases with increasing *q*, its full-width half-maximum (FWHM) rising from 7.6 ± 3.8 meV at *q* = 0.03 r.l.u. (the lowest *q* at which it can be estimated) to 46.2 ± 3.9 meV at *q* = 0.08 r.l.u. (Extended Data Fig. 7). The mode is overdamped for momenta greater than *q*c = 0.08 r.l.u., which we identify as its critical momentum. The velocity is temperature dependent, falling to 0.485 ± 0.081 eV Å at *T* = 30 K (Fig. 4a–c), and anisotropic, increasing to 0.815 ± 0.135 eV Å in the (1,1) direction (Fig. 4c).

This excitation is clearly electronic. Its velocity is approximately 100× that of the acoustic phonons, which propagate at the sound velocity, 0.008 eV Å (ref. 39). Nevertheless, its velocity is three orders of magnitude too slow to be a surface plasmon, which is gapless in the polariton regime and propagates near the speed of light[40]. The mode velocity is, however, within 10% of the velocity of the gapless mode predicted by RPA (Figs. 2b–d and 4a,b). We posit that this excitation is a demon, predicted by Pines 67 years ago but not seen in a 3D metal until now.

To check this assignment, we assess whether the mode is neutral, by examining the momentum dependence of its intensity. As illustrated in Fig. 2a, the intensity of a conventional plasmon should have the same momentum dependence as the *f*-sum rule. If the excitation is neutral, its intensity should scale with a higher power of *q*, assuring that $\varepsilon(q, 0) = 1/[1 + V(q)\chi(q, 0)] \rightarrow 1$ as *q* → 0, meaning the excitation does not contribute to screening at macroscopic distances. One complication is that M-EELS measures the response of a semi-infinite system as probed through its boundary[34], which satisfies a different sum rule than the Lindhard susceptibility computed in RPA. It is therefore crucial that we make a comparison with the correct sum rule for our experiment.

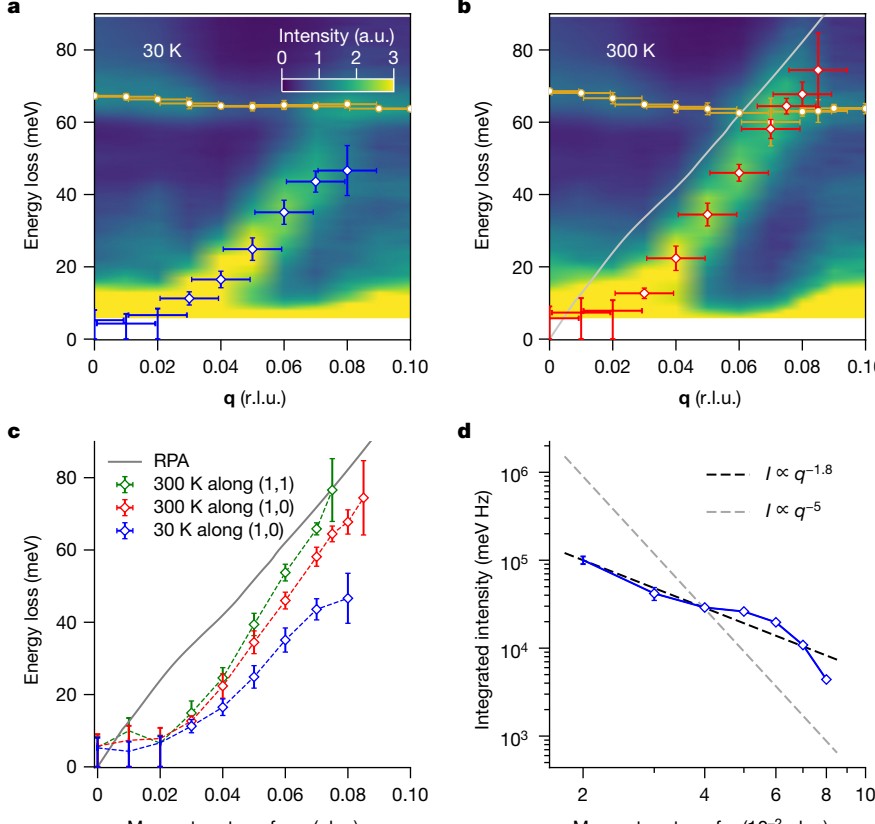

**Fig. 4 | Properties of the demon excitation in Sr₂RuO₄. a,b,** Dispersion of the demon mode in the (1,0) direction at $T = 30$ K (**a**, blue) and 300 K (**b**, red), compared with the predicted dispersion from RPA (grey). The weakly dispersing excitation at 63 meV is an optical phonon. Vertical error bars represent the fit error, whereas horizontal error bars represent the momentum resolution of the instrument (Methods). **c,** Anisotropy and temperature dependence of the demon dispersion. Horizontal error bars are omitted from this panel for clarity.

**d,** Integrated intensity of the demon excitation at $T = 30$ K (blue) as a function of $q$, showing an approximate power law $I_0(q) \approx q^{-1.8}$ (black dashed line), demonstrating that the excitation is neutral in the long-wavelength limit. For reference, the power-law scaling expected for an ordinary (charged) excitation $I_0(q) \approx q^{-5}$ is also shown (grey dashed line). We observed the demon in five distinct measurements from four different Sr₂RuO₄ crystals. a.u., arbitrary units.

The $f$-sum rule for M-EELS is derived in Methods. The result for a gapless mode is

$$I_0(q) = \frac{\hbar \sigma_0 e^2 \rho_0}{m \varepsilon_0 \alpha} \frac{1}{q^5}$$

where $q$ is the momentum and $I_0(q)$ is the energy-integrated intensity of the acoustic mode, $\hbar$ is Planck's constant, $\sigma_0$ is a cross-section scale, $\rho_0$ is the material density, $m$ is the electron mass, $\alpha$ is the dispersion coefficient, and $\varepsilon_0$ is the vacuum permittivity (see Methods). If the mode is neutral, its intensity should exhibit a power law that is higher than $q^{-5}$. The experimental intensity for the acoustic mode is shown in Fig. 4d. The best fit gives a power law $I_0(q) \approx q^{-1.83}$. This exponent is larger than −5, indicating that the excitation is neutral. We conclude that this acoustic mode is Pines' demon, predicted in 1956 but not observed in a 3D material until now.

Not every multiband metal is guaranteed to exhibit a demon. Two bands must be sufficiently different, for example by having different Fermi velocities, to give rise to a distinct pole in the charge response. Further, if Landau damping is strong, the demon may be overdamped and not visible. Nevertheless, the conditions for forming a demon are not unique to Sr₂RuO₄ and may be present in many materials.

The damping of the demon (Extended Data Fig. 7) is surprisingly small, and notably less than the scattering rate measured in infrared optics, which ranges from 20 to 50 meV, depending upon the temperature[28]. This may be due, in part, to the quasi-one-dimensional nature of

the $\beta$ band, which creates an 'eye-shaped' region in $(q,\omega)$ space in which the two-particle density of states is reduced (Extended Data Fig. 9a). The dispersion curve of the demon lies in this region, causing Landau damping to be suppressed. The neutrality of a demon also causes it to couple weakly to other excitations in the system, further enhancing its lifetime.

A demon may be thought of as a collective mode of fully screened, neutral quasiparticles or, equivalently, as a plasmon-like modulation of two different bands that, excited out of phase, leaves the total density uniform (Fig. 1b). Demons have been conjectured to mediate superconductivity and may play an important role in the low-energy physics of many multiband metals[2–7].

What enabled the current observation of the demon was meV-resolved EELS measurements using a collimated, defocused beam with high $q$ resolution. A great deal more might be learned about demons using high-energy electrons in a meV-resolved scanning transmission electron microscope (STEM) operating in an analogous, defocused configuration.

A more sophisticated theory of demons is needed. One reason is that RPA fails to predict the $q^2$ dispersion 'foot' at $q < 0.03$ r.l.u. (Fig. 4a–c), which may signify the importance of disorder, local field or excitonic effects, vertex or self-energy corrections. A full, hydrodynamic theory of demons, that properly accounts for relative motion of electrons and holes in different bands, might yield new insight into the damping mechanisms of the demon and lead to reconsideration of the role of the $\alpha$ band in this excitation.

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

## Methods

### Sample growth and characterization

Millimetre-sized, high-quality single crystals of $Sr_2RuO_4$ for M-EELS and STEM-EELS experiments were grown by a floating-zone technique reported previously[35]. Crystals were verified to have a superconducting transition temperature of about 1.5 K by alternating current susceptibility. Samples for M-EELS were cleaved in ultra-high vacuum to reveal atomically flat surfaces. A focused ion beam lamella oriented along the *ab* plane was prepared for STEM-EELS using an FEI Scios 2 focused ion beam instrument.

### M-EELS measurements

M-EELS measurements were carried out with a high-resolution EELS (HR-EELS) spectrometer modified to achieve both high momentum accuracy and precision[35] (Extended Data Fig. 4). The primary beam energy was chosen to be 50 eV, with energy and momentum resolutions of 6 meV and 0.03 Å$^{-1}$, respectively.

Single crystals of $Sr_2RuO_4$ were mounted onto oxygen-free high-conductivity copper pucks (Extended Data Fig. 1a) along with an aluminium top post using silver epoxy (EPOTEK H20-E) cured at 120 °C. Samples were cleaved at 300 K in $1.5 \times 10^{-10}$ torr vacuum and were oriented in situ based on the (0, 0) and (1, 0) Bragg reflections as observed with M-EELS at zero energy loss (Extended Data Fig. 1b). Only cleaves resulting in atomically flat surfaces and resolution-limited Bragg reflections were used for the measurements reported here. The out-of-plane momentum transfer was held fixed at $q_z = 3.95$ Å$^{-1}$ (that is, Miller index $L = 8$) throughout the entire experiment.

M-EELS spectra of the high-energy continuum were obtained by dividing out the momentum-dependent Coulomb matrix element and antisymmetrizing to remove the Bose factor[34]. It is noteworthy that, under certain conditions, neglecting the effects of the Coulomb matrix element can result in an artificially dispersing loss peak with dispersion velocity equal to the velocity of the incident probe electron (27.6 eV Å for a 50 eV electron). This artefact arises owing to the combination of geometry and the Coulomb matrix element, and only occurs when the magnitude of the probe electron's momentum perpendicular to the surface is larger after scattering (that is, backward scattering)[41]. We avoid this geometric artefact by both dividing out the Coulomb matrix element and always working in the forward-scattering geometry where the magnitude of the outgoing momentum perpendicular to the surface is smaller after scattering[34]. In any case, one should note that such geometric effects are irrelevant in the low-energy demon regime because the probe electron velocity at 50 eV is around 50 times larger than that of the plasmon.

M-EELS spectra of the high-energy continuum, shown in Fig. 3, were scaled for visibility. The spectra at different momenta were multiplied by a factor of $q^2$ and scaled so that their energy-integrated first moment is equal to that of the optical charge susceptibility in the same energy region (that is, scaled to $-\pi N_{eff}/2m$, where $N_{eff} = 3.21 \times 10^{-4}$ Å$^{-3}$ and $m$ is the free electron mass)[28]. This scaling gives the spectra units of eV$^{-1}$Å$^{-3}$.

### STEM-EELS measurements

The high-energy continuum shown in Fig. 3 of the main manuscript closely resembles that observed previously in $Bi_2Sr_2CaCu_2O_{8+x}$ (refs. 37,38), indicating it may be a generic high-energy property of strange metals. To test whether this continuum is a property of the bulk, we performed transmission EELS measurements on the same materials.

STEM-EELS measurements were performed within a Nion UltraSTEM instrument at Rutgers University with a 60 keV primary beam energy and a FWHM energy resolution of 10 meV. The angular convergence semi-angle of the beam was 30 mrad. Combined with the size of the exit aperture, these experiments probe a momentum range centred at $q = 0$ with a width $\Delta q = 5.94$ Å$^{-1} \approx 3.5$ r.l.u., so they can be considered a fully momentum-integrated measurement. STEM-EELS was performed on a single-crystal lamella of $Sr_2RuO_4$ oriented with the *ab* plane perpendicular to the incident electron beam. This lamella was lifted out and thinned down to electron transparency using an FEI Scios 2 focused ion beam instrument.

STEM-EELS spectra were acquired in a crystalline region approximately 45 nm thick ($t/\lambda \approx 0.8$ where $t$ is the sample thickness and $\lambda \approx 60$ nm is the scattering length at 60 keV) and integrated over the non-energy-dispersive direction of a 2D complementary metal–oxide–semiconductor gain-corrected image with an acceptance semi-angle of 16 mrad. From there, the momentum-integrated dynamic charge susceptibility, $\chi''(q, \omega)$, was obtained by antisymmetrizing to remove the Bose factor and then applying the same normalization as was done for M-EELS (see previous section).

A comparison between M-EELS and STEM-EELS data from $Sr_2RuO_4$ is shown in Extended Data Fig. 2a. The spectra from the two techniques are nearly identical. Although the STEM-EELS data are momentum-integrated, this comparison is meaningful because the continuum observed in M-EELS measurements is momentum independent (Fig. 3). This comparison therefore verifies the bulk nature of the high-energy continuum in $Sr_2RuO_4$.

### Surface passivation

Proper surface preparation is critical for reliable M-EELS measurements of $Sr_2RuO_4$. When cleaved in ultra-high vacuum at cryogenic temperatures, the surface of $Sr_2RuO_4$ forms dangling bonds that result in a partially filled band and a surface state whose origin is unrelated to the bulk electronic structure[26,36].

This surface state complicated the interpretation of early ARPES experiments[25,32], and could result in an extraneous 2D surface state plasmon in M-EELS measurements of the sort observed on some transition metal surfaces[19,42]. The cleaved surface of $Sr_2RuO_4$ also exhibits a $\sqrt{2} a \times \sqrt{2} a$ lattice reconstruction associated with coordinated rotation of the $RuO_6$ octahedra[43]. This superstructure results in band folding that is clearly visible in ARPES experiments[26]. Obtaining bulk-like properties in surface experiments requires suppressing both the surface state and the lattice reconstruction[26].

In ref. 36, Stöger et al. demonstrated that CO exposure passivates the surface state of ruthenium oxides by forming metal carboxylate groups that terminate the dangling surface bonds[36]. This reaction has an activation barrier of 0.17 eV, so complete passivation of the surface takes a few hours at cryogenic temperatures[26,36] and is essentially instantaneous at room temperature. CO passivation also disorders the $\sqrt{2} a \times \sqrt{2} a$ reconstruction, suppressing the surface band folding and resulting in pristine bulk bands in ARPES that match both electronic structure calculations and the observed periods in quantum oscillation experiments[26,27,35].

We therefore cleaved our surfaces at room temperature, rather than at cryogenic temperature, and then exposed them for several hours to residual CO gas with a partial pressure of $3 \times 10^{-11}$ torr—a net exposure of order approximately 0.25 Langmuir. At this exposure, the surface should be fully passivated. We confirmed that this procedure results in a disordered $\sqrt{2} a \times \sqrt{2} a$ reconstruction by measuring the (1/2,1/2) surface Bragg reflection and confirming that it is weak and highly broadened with a width $\Delta H \approx 0.2$ r.l.u. (ref. 44). In all other respects, the surface is crystallographically perfect, as demonstrated by the resolution-limited specular and (1,0) low-energy electron diffraction reflections shown in Extended Data Fig. 1b. M-EELS measurements on these surfaces should therefore be reliable and exhibit properties representative of the bulk electronic structure, as demonstrated in ref. 26.

### Anisotropy of the high-energy continuum

The band structure of $Sr_2RuO_4$ is anisotropic in the *ab* plane, as is the dispersion of the demon mode shown in Fig. 4. It is therefore important to characterize whether the high-energy continuum (Fig. 3) is similarly anisotropic. We measured the continuum at a single momentum $q = 0.5$ r.l.u. along the (1, 1) direction, that is, $(H, K) = (\frac{1}{\sqrt{2}}, \frac{1}{\sqrt{2}})$, to

compare with $q = 0.5$ r.l.u. along the $(1, 0)$ direction. These spectra are shown in Extended Data Fig. 2b. We find that the response is very similar along the two directions, indicating that the strange metal fluctuations are isotropic in-plane, despite the strong anisotropy of other aspects of the electronic structure.

## Temperature dependence of the high-energy continuum

The high-energy continuum in $Sr_2RuO_4$ is slightly temperature dependent. As shown in Extended Data Fig. 3, when the temperature is reduced from 300 K to 30 K, the continuum is slightly reduced at lower energy. This behaviour mimics that observed previously in over-doped $Bi_2Sr_2CaCu_2O_{8+x}$ (ref. 38) and is consistent with the widely held belief that, whereas $Sr_2RuO_4$ has some strange metal properties at high temperature and high-energy scales, at low temperature it is more like a Fermi liquid.

## Momentum resolution of M-EELS versus HR-EELS

Previous HR-EELS studies of $Sr_2RuO_4$ did not observe the demon mode[43] (Fig. 4). The reason for this is the difference in the momentum resolution of HR-EELS compared to M-EELS. The demon is rapidly dispersing and is only visible at momenta $q < q_c = 0.08$ r.l.u. As illustrated in Extended Data Fig. 4, the momentum resolution in ref. 43, measured by the FWHM of the specular reflection, is $0.14$ Å$^{-1} \approx 0.08$ r.l.u. This measurement therefore integrated over the entire dispersion curve of the demon. By comparison, the same measurement for our M-EELS instrument yields a resolution of 0.017 r.l.u. (Extended Data Fig. 4). This improved $q$ resolution is what allows the demon to be visible in the current measurements.

## Fitting of the demon mode

The dispersions of the acoustic demon mode and the 67 meV optical phonon shown in Fig. 4 were determined by fitting the quasi-elastic line to a pseudo-Voigt function (that is, a weighted sum of a Gaussian and Lorentzian), the acoustic mode to an antisymmetrized Lorentzian, the 67 meV optical phonon to a Fano profile (following previous work in refs. 39,43) and the 25 meV, 35 meV and 50 meV optical phonons (when present) to Lorentzians. For these fits, we focused on the raw data, that is, before dividing the matrix elements or antisymmetrizing. The error bars in Fig. 4 represent the confidence interval determined from the chi-squared value and the corresponding diagonal component of the covariance matrix from fits of this model to the experimental data. Sample fits are shown in Extended Data Fig. 5. Line plots of the demon dispersion, that is, of the data from Fig. 4, are shown in Extended Data Fig. 6.

As the dispersion of optical phonons is well documented experimentally and theoretically[39,43], we focus here on the acoustic demon mode. The FWHM of the mode is plotted in Extended Data Fig. 7, which shows that the linewidth grows with increasing momentum. Some of this width is due to the steep dispersion of the mode and the finite momentum resolution of the M-EELS measurement. However, the linewidth becomes nearly 40 meV by $q \approx 0.07$ r.l.u., indicating that intrinsic decay channels are also present. The increasing width with $q$ is most likely a consequence of Landau damping, which is commonly observed in conventional plasmons in metals[45]. For momenta $q > 0.08$ r.l.u., the mode is overdamped and no longer visible, identifying $q_c = 0.08$ r.l.u. as its critical momentum. At lower temperature, $T = 30$ K, there is a slight sharpening of the demon mode. This may be due to the reduction in the single-particle continuum shown in Extended Data Fig. 3, which could result in fewer decay channels.

For $q \le 0.02$ r.l.u., the demon mode is no longer resolvable from the tail of the quasi-elastic line due to the finite energy and momentum resolution of the experiment (Extended Data Fig. 8). The mode energy is therefore indistinguishable from zero and can be considered gapless. In this momentum region, the vertical error bars in Fig. 3 of the main

manuscript represent bounds. The value of this bound is subject to systematic errors that depend on the model used. To make an estimate of this bound, we fix the elastic line to be a Gaussian and attribute the non-Gaussian tail to the demon mode through two different schemes. In scheme A, we attribute all of this extra tail to the demon mode. In scheme B, we attribute the non-Gaussian tail to a sum of the demon and an unresolvable 'scheme B mode'. We then place the upper bound on the peak position in energy of the demon mode in Fig. 3 at the larger of the two values. At $q = 0.00$ Å$^{-1}$ (Extended Data Fig. 8) the upper bound on the demon energy gap is 8 meV.

## Multiband RPA calculations

To understand the origin of the gapless made presented in the main manuscript, Fig. 3, we calculated the collective charge modes of $Sr_2RuO_4$ using Lindhard theory in the RPA[45]. These calculations were performed without any adjustable parameters, without any optimization or fitting.

## Hamiltonian

We work with the following Hamiltonian as an effective description of the low-energy electronic degrees of freedom in $Sr_2RuO_4$.

$$H = \sum_{k,s} \mathbf{c}_s^\dagger(k) \mathbf{A}_s(k) \mathbf{c}_s(k) + \frac{1}{2} \sum_q V(q)\rho(q)\rho(-q). \tag{1}$$

Here, $\mathbf{c}_s(k) = [d_s^{yz}(k)\ d_s^{xz}(k)\ d_{-s}^{xy}(k)]^T$, where $d_\sigma^i(k)$ annihilates an electron in orbital $i$ with spin $\sigma$ and momentum $k$. Following ref. 46, we use a tight-binding band structure given by

$$\mathbf{A}_s(k) = \begin{bmatrix} \varepsilon_k^{yz} - \tilde{\mu} & \varepsilon_k^{off} + is\lambda & -s\lambda \\ \varepsilon_k^{off} - is\lambda & \varepsilon_k^{xz} - \tilde{\mu} & i\lambda \\ -s\lambda & -i\lambda & \varepsilon_k^{xy} - \tilde{\mu} \end{bmatrix}, \tag{2}$$

where

$$\begin{aligned} \varepsilon_k^{yz} &= -2\tilde{t}_2 \cos(k_x) - 2\tilde{t}_1 \cos(k_y) \\ \varepsilon_k^{xz} &= -2\tilde{t}_1 \cos(k_x) - 2\tilde{t}_2 \cos(k_y) \\ \varepsilon_k^{xy} &= -2\tilde{t}_3(\cos(k_x) + \cos(k_y)) - 4\tilde{t}_4 \cos(k_x)\cos(k_y) \\ &\quad - 2\tilde{t}_5(\cos(2k_x) + \cos(2k_y)) \\ \varepsilon_k^{off} &= -4\tilde{t}_6 \sin(k_x)\sin(k_y). \end{aligned} \tag{3–6}$$

The parameters are determined in ref. 46 by fitting to low-energy photoemission spectra. In units of electronvolts, the parameters are $\lambda = 0.032$, $\tilde{t}_1 = 0.145$, $\tilde{t}_2 = 0.016$, $\tilde{t}_3 = 0.081$, $\tilde{t}_4 = 0.039$, $\tilde{t}_5 = 0.005$, $\tilde{t}_6 = 0.000$ and $\tilde{\mu} = 0.122$. The Coulomb interaction is

$$\begin{aligned} V(q) &= \frac{e^2}{\varepsilon_0 \varepsilon_\infty} \frac{1}{q^2} \\ &= \left[ 0.31304 \text{ eV} \times \frac{a^2 c}{2} \right] \frac{1}{\left( q/\frac{2\pi}{a} \right)^2}. \end{aligned} \tag{7,8}$$

We have used lattice constants $a = 3.873$ Å and $c = 12.7323$ Å and the high-frequency dielectric constant $\varepsilon_\infty = 2.3$ from ref. 28. Here $\frac{a^2 c}{2}$ is the volume per Ru atom.

The charge density is

$$\rho(q) = \sum_{k,s} \mathbf{c}_s^\dagger(k) \mathbf{c}_s(k+q). \tag{9}$$

We approximate the charge density of each orbital as entirely localized at the centre of each Ru atom. This is a reasonable approximation for $q$ smaller than the inverse of the size of a Ru $d$ orbital.

## Band basis

To facilitate calculations, we diagonalize the non-interacting part of the Hamiltonian

$$\mathbf{c}_s^\dagger(k)\mathbf{A}_s(k)\mathbf{c}_s(k) = \sum_{a,k} c_a^\dagger(k)\varepsilon_a(k)c_a(k) \tag{10}$$

$$c_a(k) = \sum_i U_{ia}^*(k)c_i(k). \tag{11}$$

There are three bands, labelled $\alpha$, $\gamma$ and $\beta$ in order of increasing energy. Each is doubly degenerate due to pseudospin. Therefore, in the following sections, we work with one pseudospin species and restore factors of 2 as necessary.

In the band basis, the charge density can be written as

$$\rho(q) = \sum_{iab,k} U_{ia}^*(k)U_{ib}(k+q)c_a^\dagger(k)c_b(k+q). \tag{12}$$

Therefore, the total density can be decomposed as

$$\rho(q) = \sum_{ab} \rho_{ab}(q) \tag{13}$$

$$\rho_{ab}(q) = \sum_{i,k} U_{ia}^*(k)U_{ib}(k+q)c_a^\dagger(k)c_b(k+q). \tag{14}$$

The density operator involves both band densities (for example, $c_\alpha^\dagger c_\alpha$) and interband excitations (for example, $c_\alpha^\dagger c_\beta$). This decomposition will be useful later in analysing partial susceptibilities.

## Charge susceptibility

The non-interacting charge susceptibility is

$$\chi^0(q,\omega) = \frac{2}{N}\sum_{ijab,k} U_{ia}(k+q)U_{ja}^*(k+q)U_{jb}(k)U_{ib}^*(k)$$
$$\frac{f(\varepsilon_b(k)) - f(\varepsilon_a(k+q))}{\omega + \varepsilon_b(k) - \varepsilon_a(k+q) + i0^+}. \tag{15}$$

Here, $i,j$ are orbital indices and $a,b$ are band indices. $N$ is the number of $k$-points summed over and $f(\varepsilon) = (e^{\varepsilon/T}+1)^{-1}$ is the Fermi–Dirac function. In figures showing the demon, we use a 1,000 × 1,000 grid of $k$-points uniformly distributed over the first Brillouin zone. The temperature is set to 30 K and a small Lorentzian broadening of $\gamma = 3$ meV is applied through substituting $i0^+ \to i\gamma$. In figures showing the plasmon, we use a 400 × 400 grid of $k$-points and a Lorentzian broadening of $\gamma = 10$ meV. A plot of $-\mathrm{Im}\chi^0(q,\omega)$ is shown in Extended Data Fig. 9a. The features seen here may be understood through the band decomposition described in the next section.

Under the RPA, the full charge susceptibility is given by

$$\chi(q,\omega) = \frac{\chi^0(q,\omega)}{1 - V(q)\chi^0(q,\omega)}. \tag{16}$$

The result is plotted in Figs. 2a,b and in Extended Data Fig. 9b.

Interestingly, looking closely at Extended Data Fig. 9b reveals an additional excitation at $\omega \approx 20$ meV, which appears as a shoulder on the demon excitation. It is likely that this extra peak is a second demon, owing to the interaction between the $\alpha$ and $\gamma$ bands. This $\alpha$–$\gamma$ feature is at lower energy and contains less spectral weight than the primary, $\beta$–$\gamma$ demon, because of the much smaller Fermi surface volume of $\alpha$ band. We therefore did not see it in our experiments. Future measurements with better resolution might reveal this additional feature.

## Demon intensity

The imaginary part of the total charge susceptibility calculated by RPA is plotted against frequency in Extended Data Fig. 9b at small $q$. The linearly dispersing demon is the most prominent feature at these momenta. Its peak intensity scales approximately as $q^4$. Given that the peak width also increases with $q$, the demon clearly fails to satisfy any partial $f$-sum rule, consistent with expectations for a neutral excitation (see main text and the 'Sum rule' section below).

In Extended Data Fig. 9b, a second mode is visible as well at lower energies (for instance, 30 meV for $q = (0.1, 0)$). As this mode is also linearly dispersing with intensity scaling $q^4$, we identify the mode as a second demon involving the $\alpha$ and $\gamma$ bands. Unlike the primary demon, this mode is strongly Landau damped due to the considerable intensity of the particle–hole continuum in Extended Data Fig. 9a.

## Band decomposition of the susceptibility

The susceptibility describes the response of the total charge density to a potential that couples to the total charge density. As charge density can be decomposed into components in equation 13, we define a susceptibility matrix, $\chi(q,\omega)$, where each element describes the response of a component of the charge density to a potential that couples to a single component. To be precise,

$$\chi_{ab,cd}(q, i\omega_n)$$
$$= -\frac{1}{N}\int_0^\beta d\tau e^{i\omega_n\tau}(\langle\rho_{ab}(q,\tau)\rho_{cd}(-q)\rangle - \langle\rho_{ab}(q)\rho_{cd}(-q)\rangle). \tag{17}$$

The susceptibility follows after analytically continuing $i\omega_n \to \omega + i0^+$. The non-interacting result is

$$\chi_{ab,cd}^0(q,\omega) = \delta_{ad}\delta_{bc}\frac{2}{N}\sum_{ij,k} U_{ia}(k+q)U_{ja}^*(k+q)U_{jb}(k)U_{ib}^*(k)$$
$$\frac{f(\varepsilon_b(k)) - f(\varepsilon_a(k+q))}{\omega + \varepsilon_b(k) - \varepsilon_a(k+q) + i0^+}. \tag{18}$$

The delta functions are due to the decoupling of bands in a non-interacting system. For instance, if $a \neq d$, $\langle c_a^\dagger c_b c_c^\dagger c_d\rangle = \langle c_a^\dagger c_b\rangle\langle c_c^\dagger c_d\rangle$, so $\chi_{ab,cd} = 0$. In an interacting system, this is no longer true and all 9 × 9 elements of $\chi_{ab,cd}$ are non-zero in general.

The nine non-zero elements of $\chi^0(q,\omega)$ are plotted in Extended Data Fig. 10a. From this we can identify features as either intraband or interband excitations. At small $q$, interband transitions have an intensity of approximately $q^2$ in $\chi^0(q,\omega)$ and therefore intraband particle–hole excitations dominate. As can be seen in Extended Data Fig. 10a, the strongest contributors to $\chi^0$ are $\chi_{\gamma\gamma,\gamma\gamma}^0$ and $\chi_{\beta\beta,\beta\beta}^0$. The two bands clearly have different velocities. Importantly, at small $q$, $\mathrm{Im}\chi_{\beta\beta,\beta\beta}^0$ has spectral weight restricted to a small window of frequencies. This is due to the quasi-one-dimensional nature of the $\beta$ band. The consequence is that there is a pocket in $\mathrm{Im}\chi^0(q,\omega)$ from $q = (0,0)$ to $q \approx (0.13, 0)$ with suppressed spectral weight (Extended Data Fig. 9a). It is precisely in this pocket that the demon disperses (Fig. 2c) without becoming overdamped.

The interaction $V(q)\rho(q)\rho(-q)$ may be written as

$$V(q)\rho(q)\rho(-q) = \sum_{abcd} V_{ab,cd}(q)\rho_{ab}(q)\rho_{cd}(-q), \tag{19}$$

where $V_{ab,cd}(q) = V(q)$ for all $a, b, c$ and $d$. Therefore, we define the 9 × 9 interaction matrix $\mathbf{V}(q)$ with every element equal to $V(q)$.

Under the RPA, the matrix susceptibility is

$$\chi(q,\omega) = \chi^0(q,\omega) + \chi^0(q,\omega)\mathbf{V}(q)\chi^0(q,\omega) + \cdots$$
$$= \chi^0(q,\omega)[\mathbf{I} - \mathbf{V}(q)\chi^0(q,\omega)]^{-1}, \tag{20,21}$$

where $\mathbf{I}$ is the identity matrix and multiplication and inversion are matrix operations. It is straightforward to show that the sum of all elements in the RPA susceptibility matrix equals the scalar RPA result in equation 16.

Density–density components of the susceptibility matrix ($\chi_{aa,bb}$) may be used to determine the identity of modes in $\chi(q, \omega)$. $\chi_{aa,bb}(q, \omega)$ describes the response of the density in band $a$ to a potential that couples to the density of band $b$. These components are plotted at high frequency in Extended Data Fig. 10b and at low frequency in Extended Data Fig. 10c. Some of these components were plotted previously in Fig. 2, where we relabelled $\chi_{aa,bb} \equiv \chi_{a,b}$ for brevity.

At high frequencies (Extended Data Fig. 10b), the plasmon is visible in all density–density components. Every component has the same sign, indicating that a potential modulated at the plasmon frequency induces an in-phase oscillation of the density in all three bands. By contrast, at low frequencies (Extended Data Fig. 10c), a number of features are present including remnants of the particle–hole continua (Extended Data Fig. 9a) and the demon. The demon is visible most clearly in the elements $\chi_{\gamma\gamma,\gamma\gamma}$, $\chi_{\beta\beta,\beta\beta}$, $\chi_{\gamma\gamma,\beta\beta}$ and $\chi_{\beta\beta,\gamma\gamma}$. The sign of the susceptibility of the demon excitation in the diagonal elements, $\chi_{\gamma\gamma,\gamma\gamma}$ and $\chi_{\beta\beta,\beta\beta}$, is opposite to that of the off-diagonal elements, $\chi_{\gamma\gamma,\beta\beta}$ and $\chi_{\beta\beta,\gamma\gamma}$. This demonstrates the out-of-phase character of the demon. A potential coupling to the $\beta$ band that is modulated at the frequency of the demon excites opposite density modulations in the $\gamma$ and $\beta$ bands. This identifies the gapless mode in Extended Data Fig. 9b as a true demon that, to leading order, does not modulate the total density.

## Sum rule for surface EELS and neutrality of the demon

A demon has two defining properties. The first is that it is gapless, that is, its energy tends toward zero as $q \to 0$. The second is that it is neutral, that is, it cannot screen charge in the $q \to 0$ limit. The former property is a consequence of the latter, which eliminates the Coulomb contribution to the energy of the mode in the long-wavelength limit. Figure 4 demonstrates that the collective mode is gapless. Here we show that it is also neutral and therefore it satisfies all of the criteria for being a demon.

We can establish experimentally whether the excitation is neutral by examining the momentum dependence of its intensity. The dielectric function of a material is related to its charge susceptibility, $\chi(q, \omega)$, by

$$\epsilon(q, \omega) = \frac{1}{1 + V(q)\chi(q, \omega)} \tag{22}$$

where $V(q) = e^2/\varepsilon_0 q^2$ is the 3D Coulomb interaction. The imaginary part of the susceptibility satisfies the $f$-sum rule,

$$\int_0^\infty \chi''(q, \omega)\omega d\omega = \frac{\pi n q^2}{2m}. \tag{23}$$

In conventional metals, the spectral weight in the plasmon takes up all the weight in this sum rule and the intensity of the plasmon approximately $q^2$ at small $q$ (see, for example, Fig. 1 in ref. 47). This behaviour assures that $V(q)\chi(q, \omega)|_{\omega=0}$ converges to a constant at small $q$, allowing the material to exhibit a finite screening strength.

In the RPA calculation described above (summarized in Fig. 2), the spectral weight in the demon is a faster function of $q$ than the total spectral weight defined by the $f$-sum rule, that is, $\chi \approx q^\alpha$, where $\alpha > 2$ ($\alpha = 4$ in the RPA case). Hence, for a demon excitation, $V(q)\chi(q, \omega)|_{\omega=0} \to 0$ as $q \to 0$, so $\varepsilon \to 1$ and a demon does not contribute to screening in the long-wavelength limit. This is what is meant by the statement that a demon is 'neutral'. Determining whether the gapless mode in Fig. 4 is neutral therefore requires comparing the $q$ dependence of its spectral weight to expectations from the $f$-sum rule.

A complication is that M-EELS is a surface probe and does not measure the simple, bulk susceptibility, $\chi(q, \omega)$. M-EELS measures a surface

response function, $\chi_s(q, \omega)$, as described in detail in refs. 34,48. This surface quantity does not satisfy the same sum rule as equation 23 above. We therefore need to derive a sum rule for the response function measured with surface M-EELS and compare the $q$ dependence of the spectral weight in the excitation to this sum rule.

## Sum rule for surface M-EELS

In general, the charge susceptibility can be written as

$$\chi(\mathbf{k}, \mathbf{k}', \omega) = \sum_n \left\{ \frac{\langle 0|\hat{\rho}_\mathbf{k}|n\rangle\langle n|\hat{\rho}_{-\mathbf{k}'}|0\rangle}{\omega - \omega_{n0} + i0_+} - \frac{\langle n|\hat{\rho}_{-\mathbf{k}'}|0\rangle\langle 0|\hat{\rho}_\mathbf{k}|n\rangle}{\omega + \omega_{n0} + i0_+} \right\} \tag{24}$$

where $\hat{\rho}_\mathbf{k}$ is the charge density operator. In systems with translational symmetry, the only non-zero matrix elements of $\chi(\mathbf{k}, \mathbf{k}', \omega)$ satisfy $\mathbf{k} = \mathbf{k}' + \mathbf{G}$, where $\mathbf{G}$ is a reciprocal lattice vector. In metals, where the system is homogeneous, $\mathbf{G} = 0$. In systems that lack translational symmetry, the $f$-sum rule can be generalized to[45]

$$\int_{-\infty}^\infty d\omega\omega\chi(\mathbf{k}, \mathbf{k}', \omega) = i\pi\langle 0|[[H, \hat{\rho}_{-\mathbf{k}}], \hat{\rho}_{\mathbf{k}'}]|0\rangle. \tag{25}$$

The exact Hamiltonian $H$ can be generically expressed in terms of the kinetic energy of free electrons, which is Galilean invariant, plus potentials that depend on charge density operators. In the absence of potentials that depend explicitly on momentum operators,

$$\langle 0|[[H, \hat{\rho}_{-\mathbf{k}}], \hat{\rho}_{\mathbf{k}'}]|0\rangle = -\frac{\hbar^2}{m}\mathbf{k} \cdot \mathbf{k}'\rho_{\mathbf{k}'-\mathbf{k}}. \tag{26}$$

The generalized $f$-sum rule then becomes

$$\int_{-\infty}^\infty d\omega\omega\chi(\mathbf{k}, \mathbf{k}', \omega) = -i\pi\frac{\hbar^2}{m}\mathbf{k} \cdot \mathbf{k}'\rho_{\mathbf{k}'-\mathbf{k}}. \tag{27}$$

We now wish to apply this sum rule to experimental M-EELS data. The M-EELS cross-section is given by[34,48]

$$\frac{\partial^2\sigma}{\partial\Omega\partial E} = \sigma_0 V_{eff}^2(\mathbf{q})\int_{-\infty}^0 dz_2 dz_2 e^{-|\mathbf{q}||z_1+z_2|} \cdot S(\mathbf{q}, z_1, z_2, \omega) \tag{28}$$

where $S$ is the density–density correlation function, which is related to the density response function by the fluctuation-dissipation theorem,

$$S(\mathbf{q}, z_1, z_2, \omega) = -\frac{1}{\pi}\frac{1}{1 - e^{-\hbar\omega/k_B T}}\chi''(\mathbf{q}, z_1, z_2, \omega), \tag{29}$$

The Coulomb matrix elements

$$V_{eff}(k_z^i + k_z^s, q) = \frac{e^2/\varepsilon_0}{(k_z^i + k_z^s)^2 + q^2} \tag{30}$$

describe the coupling of the probe electron to the valence electrons near a surface, accounting for a single reflectivity event[34,48].

In a semi-infinite stack of metallic layers, translational symmetry is satisfied along the directions parallel to the metallic layers, but not in the direction perpendicular to the surface. The susceptibility has the general form $\chi(\mathbf{q}, \mathbf{q}, k_z, k_z')$, where $\mathbf{q}$ is the momentum parallel to the surface and $k_z$, $k_z'$ the momenta along the direction perpendicular to the surface. Fourier transforming equation 27 in $k_z$ and $k_z'$, the generalized $f$-sum rule can be equivalently written as

$$\int_{-\infty}^\infty p d\omega\omega\chi(\mathbf{q}, \mathbf{q}, z, z', \omega)$$
$$= -i\pi\frac{\hbar^2}{m}\left[\delta(z-z')q^2 - \frac{\partial^2\delta(z-z')}{\partial(z-z')^2} + \frac{\partial\delta(z-z')}{\partial(z-z')}\frac{\partial}{\partial z'}\right]\rho(z'), \tag{31}$$

where because of the surface $\rho(z) = 0$ for $z > 0$. Combining the scattering cross-section of M-EELS[34,48],

$$\frac{\partial^2 \sigma}{\partial \Omega \partial E}(\mathbf{q}, k_z^i, k_z^s, \omega) = -\frac{1}{\pi}\frac{1}{1-\mathrm{e}^{-\beta\omega}}\sigma_0[V_{\mathrm{eff}}(k_z^i + k_z^s, \mathbf{q})]^2$$

$$\int_{-\infty}^{0} \mathrm{d}z_1, \mathrm{d}z_2 \mathrm{e}^{q(z_1+z_2)}\mathrm{Im}\chi(\mathbf{q}, \mathbf{q}, z_1, z_2, \omega), \qquad (32)$$

with the $f$-sum rule equation 31, the sum rule for the M-EELS cross-section is

$$\int_{-\infty}^{\infty} \mathrm{d}\omega \omega (1 - \mathrm{e}^{-\beta\omega})\frac{\partial^2 \sigma}{\partial \Omega \partial E}(\mathbf{q}, k_z^i, k_z^s, \omega)$$

$$= \frac{2\hbar^2 q^2}{m}\sigma_0[V_{\mathrm{eff}}(k_z^i + k_z^s, \mathbf{q})]^2 \int_{-\infty}^{0} \mathrm{d}z\rho(z)\mathrm{e}^{2qz}. \qquad (33)$$

## Neutrality test of the collective mode

Equation 33 is written in terms of the experimental cross-section and therefore may be applied directly to the experimental data. We start by making some simplifying assumptions that apply in the small $q$ regime. The first is that the density $\rho(z) = \rho_0 \theta(-z)$, that is,

$$\int_{-\infty}^{0} \mathrm{d}z\rho(z)\mathrm{e}^{2qz} = \frac{\rho_0}{2q} \qquad (34)$$

This expression is valid as long as the width of the surface (that is, the distance over which the density falls to zero) is much less than $q^{-1}$. Next, we take $T = 0$, which for data taken at $T = 30$ K is valid for $\omega > 2.5$ meV. Finally, we need to consider the actual behaviour of the mode in the small $q$ regime. Although the mode disperses linearly over most of its range, in the small $q$ limit $E(q) \approx q^2$. We therefore take the experimental intensity to have the form

$$I(q, \omega) = I_0(q)\delta(\omega - \alpha q^2) \qquad (35)$$

where $I_0(q)$ then represents the $\omega$-integrated intensity of the mode at momentum $q$. Evaluating equation 33 then gives

$$I_0(q) = \frac{\hbar^2 \sigma_0 e^2 \rho_0}{m\epsilon_0 \alpha}\frac{1}{q^5}. \qquad (36)$$

In other words, if a collective mode encompasses all the spectral weight in the $f$-sum rule, its integrated intensity should satisfy equation 36. If, however, a mode is neutral, its spectral weight should scale with a higher power of $q$. Therefore, for a given excitation, $I_0(q) \approx q^\alpha$ in the small $q$ limit. If the excitation is neutral, then $\alpha > -5$.

We carried out this test on the gapless excitation observed with M-EELS in Fig. 4. The result is shown in Fig. 4d. The integrated intensity of the mode follows a power law of roughly $I_0(q) \approx q^{-1.8}$. Because $-1.8 > -5$, we conclude that this excitation is neutral in the sense that it cannot contribute to screening in the small $q$ limit, and therefore is a demon in the true sense.

## Data availability
The data reported in this paper have been deposited on Zenodo (available at https://zenodo.org/record/7812299).

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

**Acknowledgements** We acknowledge J. Zaanen, D. van der Marel, A. Georges, M. Zingl, H. Strand, P. Coleman, F. Flicker and J. Fink for helpful discussions. This work was primarily supported by the Center for Quantum Sensing and Quantum Materials, an Energy Frontier Research Center funded by the US Department of Energy (DOE), Office of Science, Basic Energy Sciences (BES), under award no. DE-SC0001238 (A.A.A., E.W.H., X.G., T.C.C., P.W.P. and P.A.). Growth of Sr$_2$RuO$_4$ crystals (Y.M.) was supported by JSPS grant nos. JPJSCCA20170002 and JP22H01168. Derivation of the sum rule was partially supported (B.U.) by NSF grant no. DMR-2024864. STEM-EELS measurements (P.E.B.) were partially supported by DOE grant no. DE-SC0005132. P.A. gratefully acknowledges additional support from the EPiQS programme of the Gordon and Betty Moore Foundation, grant no. GBMF9452. E.W.H. acknowledges support from EPiQS grants no. GBMF4305 and GBMF8691. M.M. acknowledges support from the Alexander von Humboldt foundation.

**Author contributions** A.A.H., M.M. and P.A. conceived the experiment. A.A.H. and M.M. performed the M-EELS experiments with support from S.I.R. and M.S.R. Together with H.Y. and P.E.B., A.A.H. also carried out STEM-EELS measurements. A.A.H. analysed the data with input from M.M., B.U., T.C.C. and P.A. Samples were grown and characterized by C.S. and Y.M. RPA calculations were provided by E.W.H. and P.W.P. B.U., X.G. and P.A. derived the M-EELS sum rule and the neutrality tests. A.A.H., E.W.H., P.W.P. and P.A. wrote the manuscript with input from all authors.

**Competing interests** The authors declare no competing interests.

**Additional information**
**Correspondence and requests for materials** should be addressed to Ali A. Husain or Peter Abbamonte.

## A

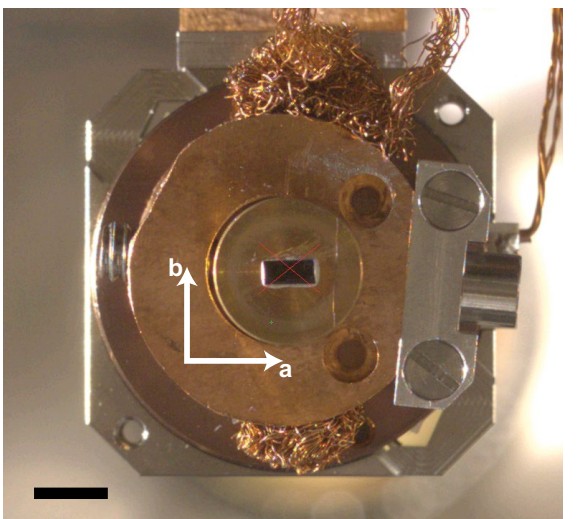

## B

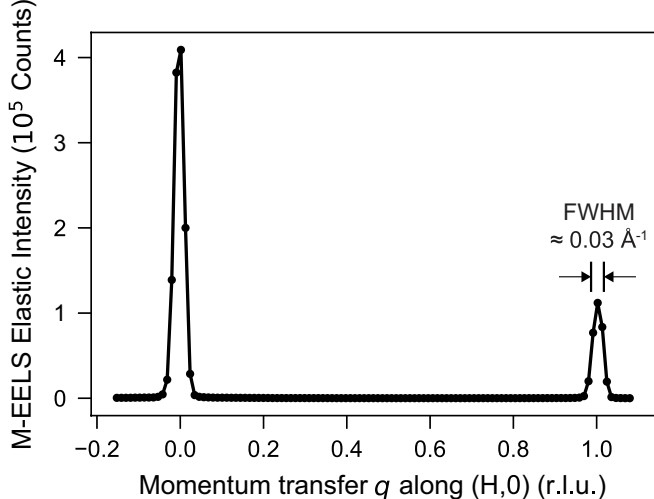

**Extended Data Fig. 1 | Cleaved Sr$_2$RuO$_4$ single crystal measured with M-EELS.**
**a**, Example Sr$_2$RuO$_4$ single crystal measured with M-EELS. The sample is mounted on an OFHC copper puck and cleaved in UHV to reveal a flat surface (scale bar 5 mm). To accurately align the instrument momentum transfer with the crystal axes, the sample is rotated azimuthally via a piezorotator. In this case the *a*-axis was aligned to be in the scattering plane. **b**, Momentum-dependence of the elastic M-EELS response, which corresponds to Bragg diffraction, of Sr$_2$RuO$_4$ along (*H*, 0). A sharp specular peak at (0, 0) is visible, as is a (1, 0) low-energy electron diffraction reflection with a FWHM of approximately 0.03 Å$^{-1}$, indicating a clean, well ordered surface.

**a**

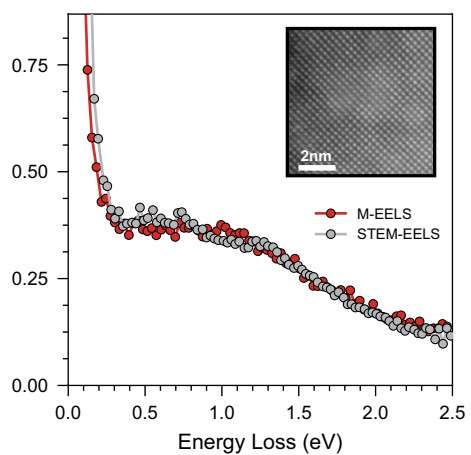

**b**

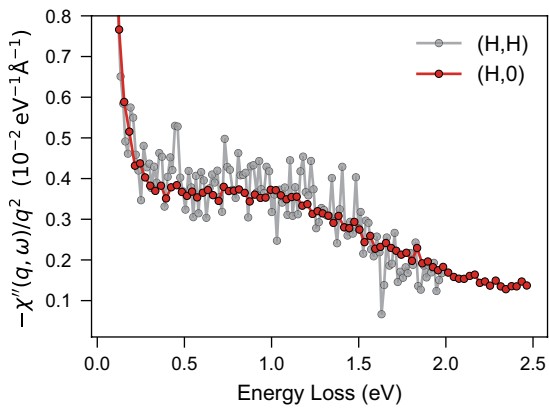

**Extended Data Fig. 2 | Properties of the high-energy excitations in Sr₂RuO₄.**
**a**, Comparison between surface M-EELS and bulk-sensitive EELS measurements of Sr₂RuO₄ with a scanning transmission electron microscope (STEM). The similarity of the two spectra verifies the bulk origin of the high-energy continuum. (Inset) high-angle annular dark field image of the sample region used for STEM-EELS measurements confirms its crystallinity. **b**, Comparison of the M-EELS response at $q = 0.5$ r.l.u. along the $(H, 0)$ and $(H, H)$ directions. To within the statistical uncertainty of the data, the overall shape of the strange metal continuum is the same in the two directions, suggesting that the high-energy continuum is roughly isotropic in Sr₂RuO₄.

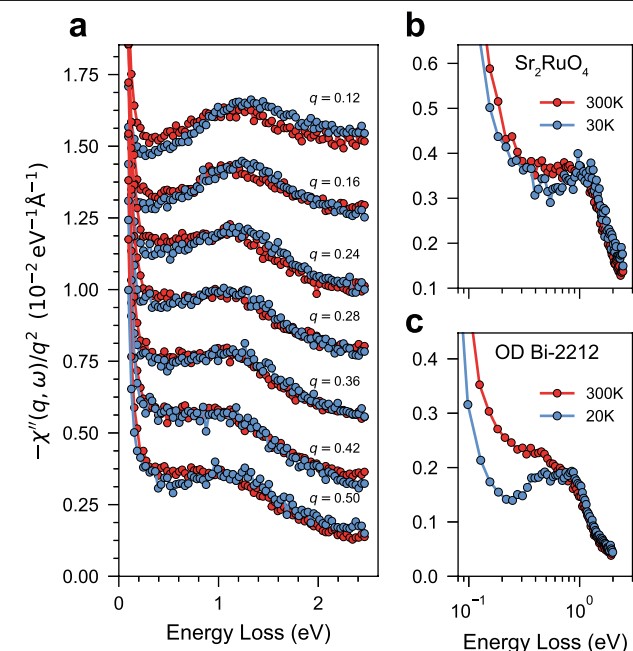

**Extended Data Fig. 3 | Temperature dependence of the high energy continuum in Sr₂RuO₄.** The spectral weight at low energy is slightly reduced at low temperature, exhibiting the same behavior as overdoped Bi₂Sr₂CaCu₂O₈₊ₓ[37,38].

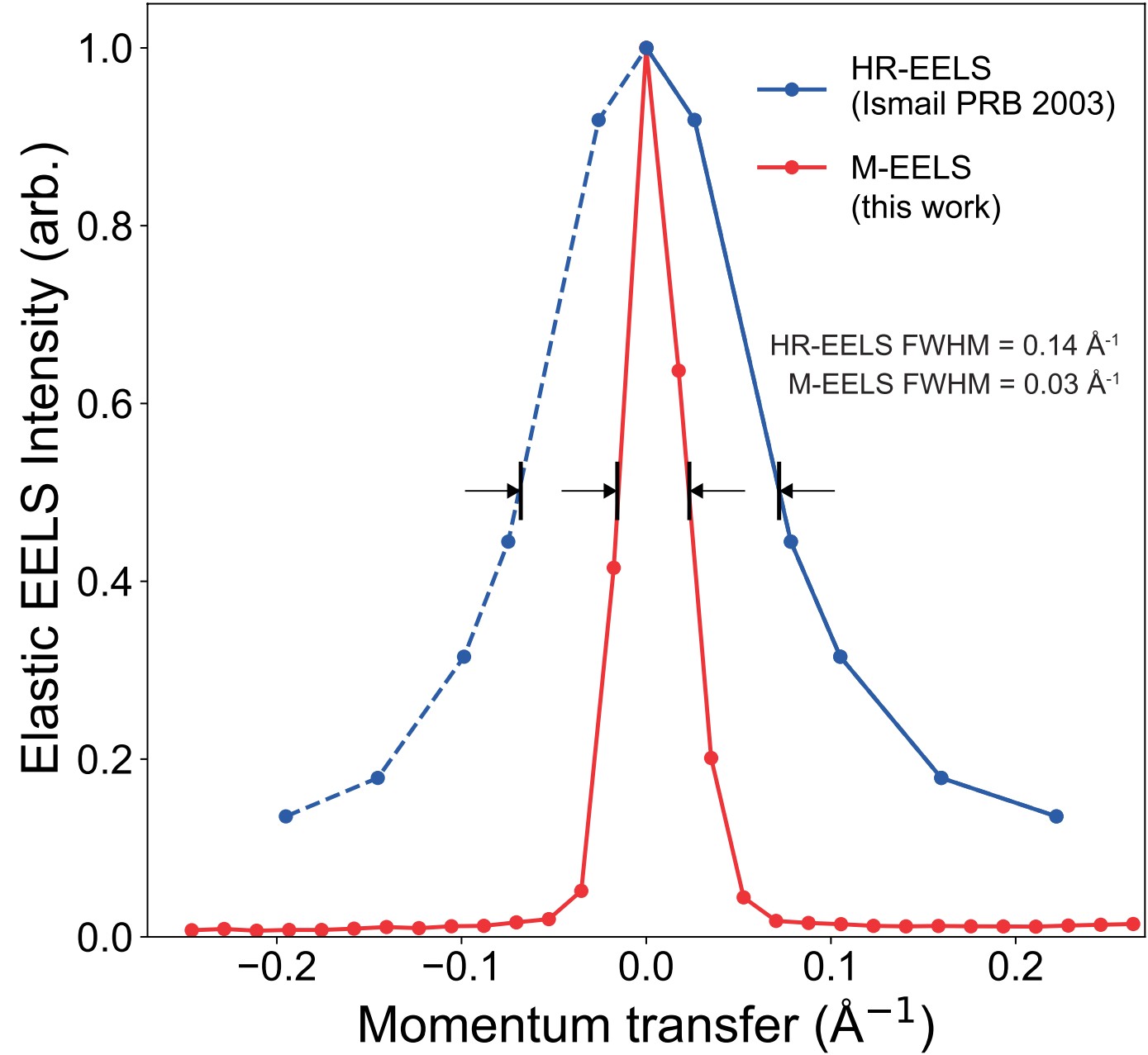

**Extended Data Fig. 4 | Comparison of the momentum resolution of HR-EELS vs. M-EELS.** Plot of the elastic specular reflection from the surface of $Sr_2RuO_4$ as a function of momentum transfer for HR-EELS from ref. 43. (blue), and for M-EELS from this work (red). The HR-EELS data were mirrored (dashed line) to obtain the FWHM since ref. 43. presented only positive values of $q$. The full-width at half-maximum of the specular reflection for M-EELS is about 0.03 Å$^{-1}$, which is nearly five times sharper than that of HR-EELS (0.14 Å$^{-1}$), despite working at a significantly higher beam energy (50 eV compared to 20 eV). Because the demon only exists below about 0.13 Å$^{-1}$ = 0.08 r.l.u., it was not visible in prior HR-EELS measurements.

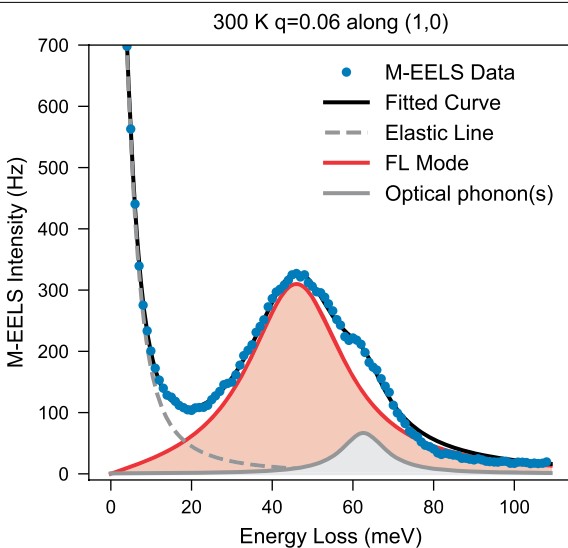

**Extended Data Fig. 5 | Example fits of the phonon and demon modes.** Three example fits of the M-EELS spectra for **a**, $q = 0.03$ r.l.u. along $(1, 1)$ at 300 K, **b**, $q = 0.06$ r.l.u. along $(1,0)$ at 300 K, and **c**, $q = 0.08$ r.l.u. along $(1, 0)$ at 30 K. Fits comprise a quasi-elastic line (grey dashed line) of pseudo-Voigt form, a demon mode using a Lorentz oscillator (red), and optical phonons (grey full lines) each with a Fano or Lorentzian line shape (see text).

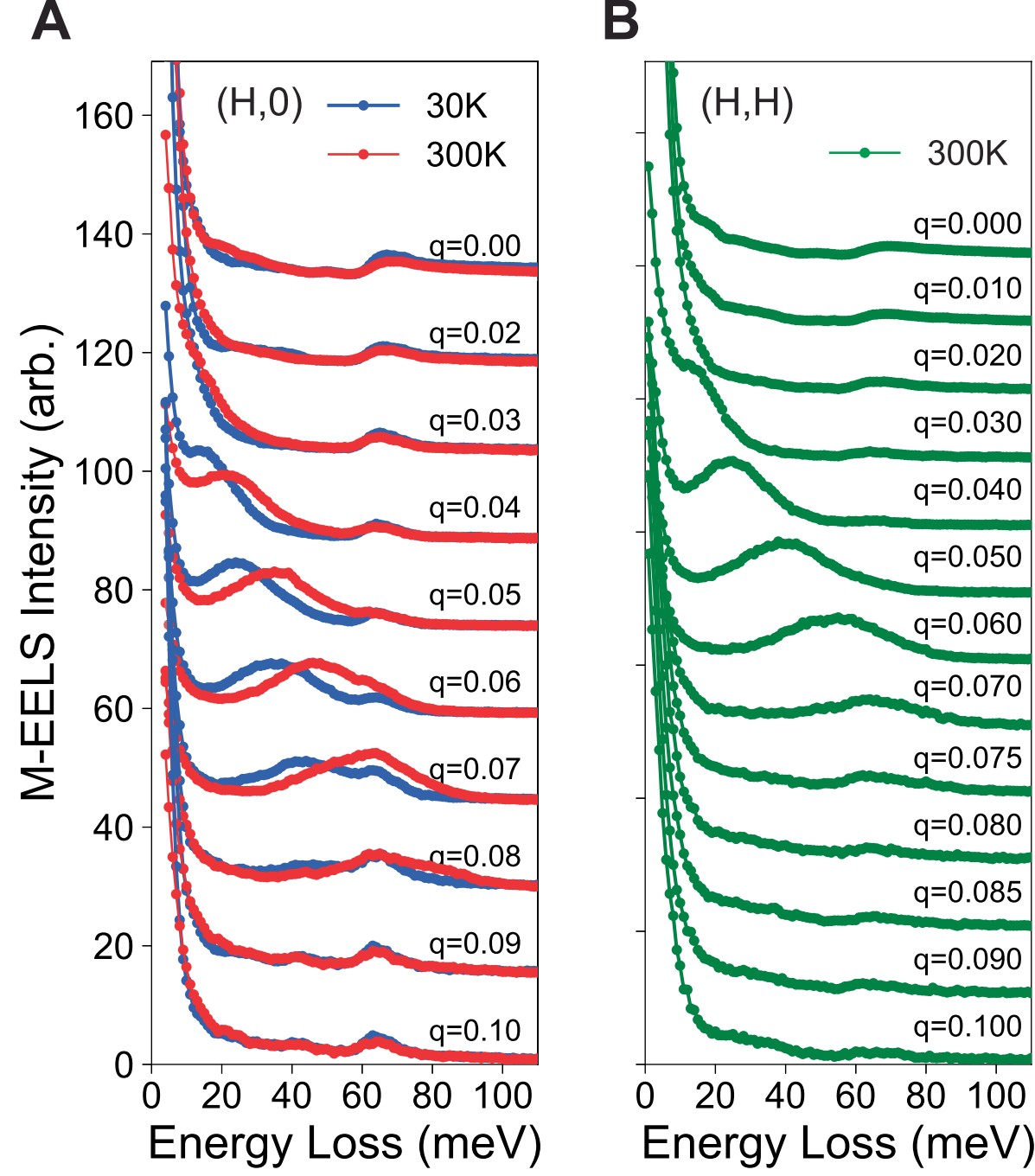

**Extended Data Fig. 6 | Line plots of the dispersion of the demon mode.** Line plots of the M-EELS spectra from Fig. 4 of the main manuscript, showing the dispersion of the demon mode along **a**, (1, 0) at 30 K (blue), 300 K (red) and **b**, along (1, 1) at 300 K (green). Spectra are offset vertically and normalized to their values at 85 meV for clarity.

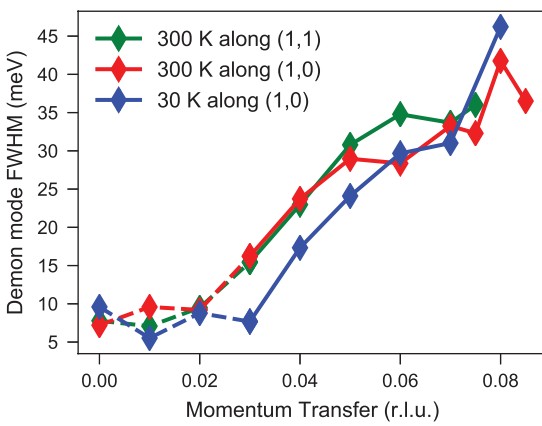

**Extended Data Fig. 7 | Width of the demon mode.** Full-width at half-maximum energy width of the demon mode as a function of momentum, $q$. The width ranges from around 8 meV at 0.03 r.l.u. to more than 40 meV at 0.08 r.l.u.

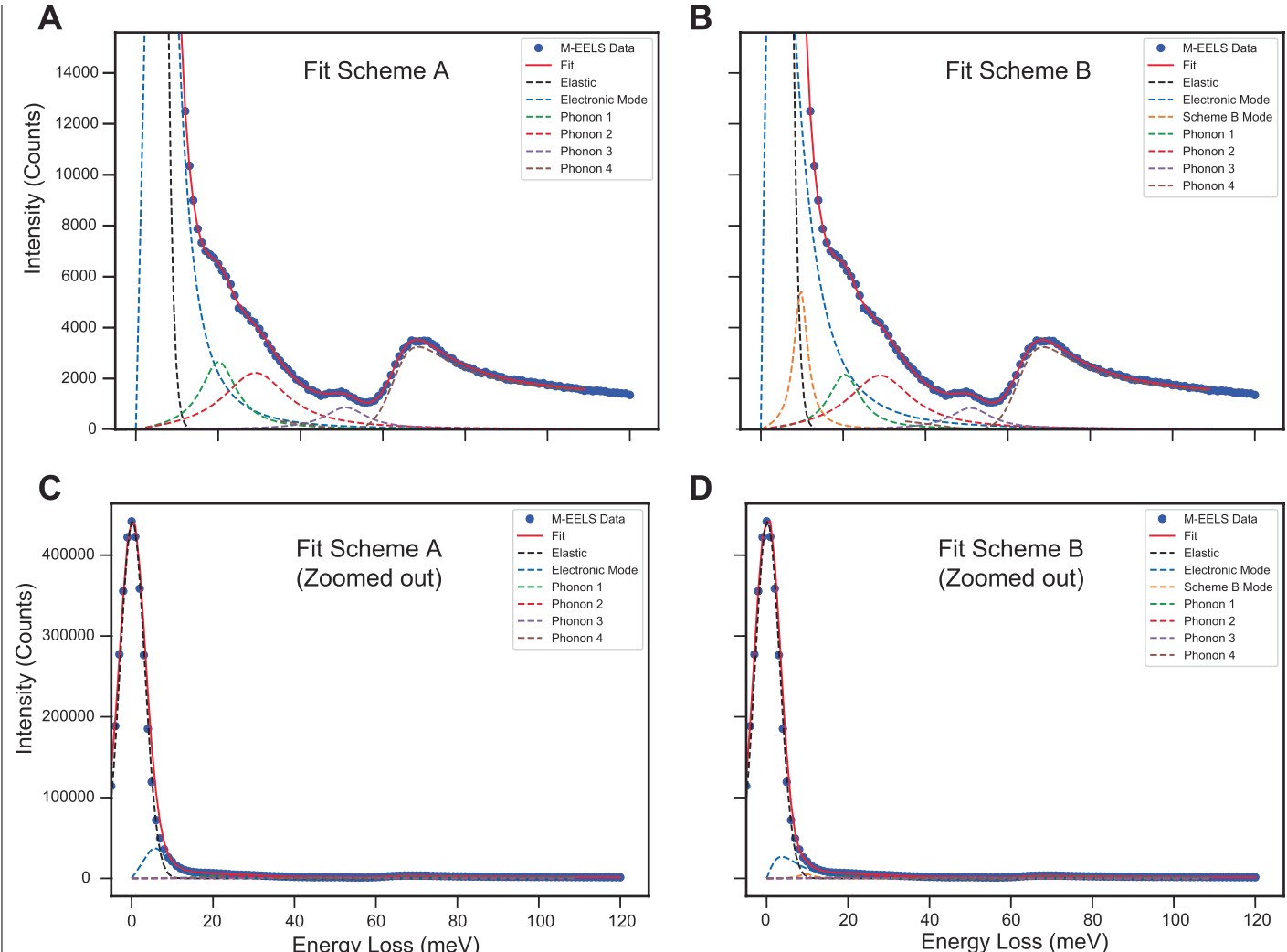

**Extended Data Fig. 8 | Upper bound on the demon mode energy at small momentum.** For M-EELS spectra at $q = 0$ r.l.u. at $T = 300$ K, the demon mode is not clearly resolvable from the elastic line. To estimate an upper bound on its energy for $q < 0.02$ Å$^{-1}$, the quasi-elastic line is fitted with a Gaussian, and the tails attributed to the demon mode in two schemes. In scheme A this tail is completely attributed to the demon, while in scheme B it is attributed to the sum of the demon and some other unresolvable mode with Lorentzian form. **a**, Fit of the spectra according to scheme A as described in the text. **b**, Fit of the same spectra as (A) but according to scheme B. **c**, Same plot as (A) but the vertical axis is zoomed out to show the quasi-elastic line and its tails. **d**, Same plot as (B) but again zoomed out vertically.

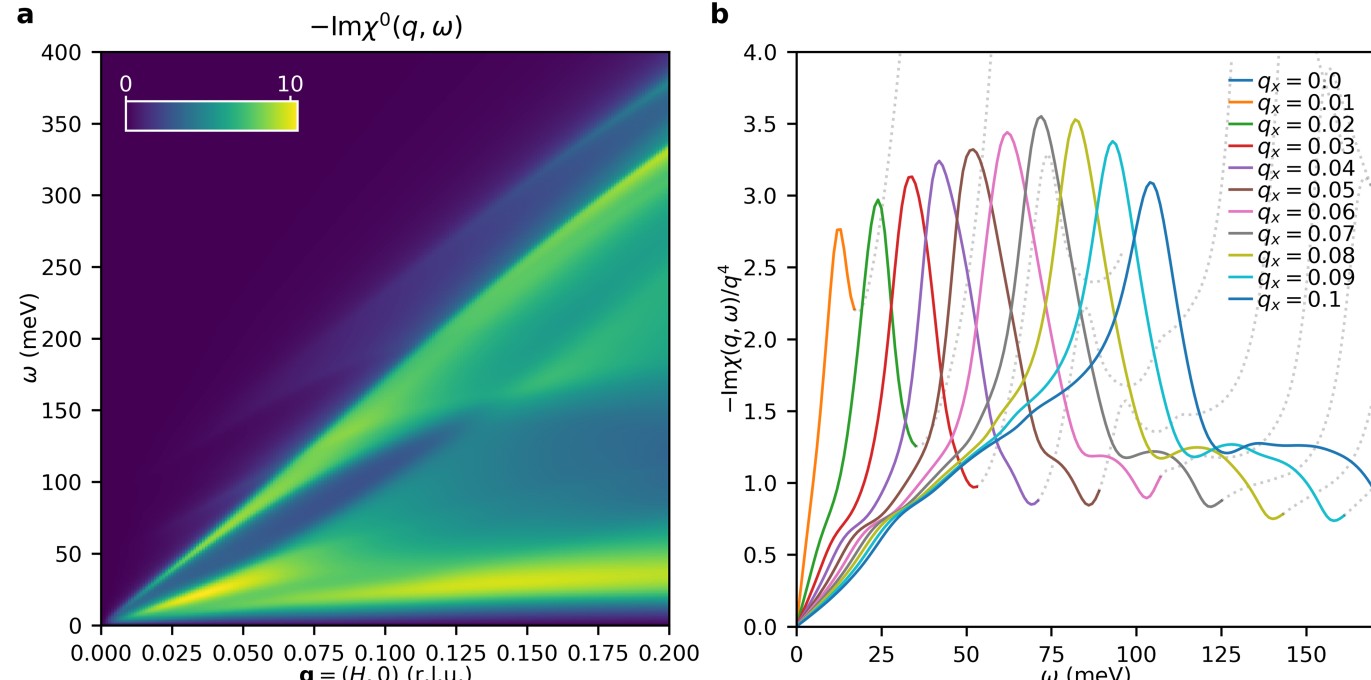

**Extended Data Fig. 9 | Low-energy excitations in Sr$_2$RuO$_4$ calculated in the random phase approximation. a**, Noninteracting particle-hole continuum of Sr$_2$RuO$_4$, i.e., the negative of the imaginary part of $\chi^0(q, \omega)$ calculated using Eq. 15. Notice that there is an 'eye-shaped' quiet spot in the spectrum, which is a consequence of the quasi-1D character of the $\beta$ band. Landau damping of the demon should be reduced in this region, enhancing its stability. **b**, Negative of the imaginary part of the full, interacting charge susceptibility at small momenta. Each spectrum is divided by $q^4$ to highlight the demon.

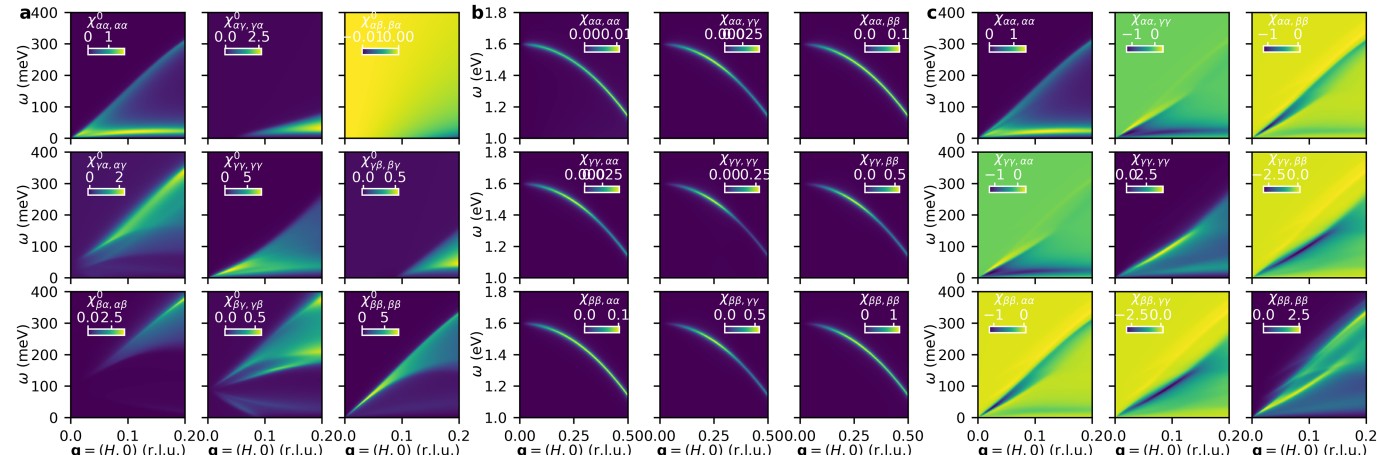

**Extended Data Fig. 10 | Band decomposition of the susceptibility matrix.**
**a**, All nonzero elements of the non-interacting susceptibility matrix. Diagonal elements show the particle-hole continua of the bands. Off-diagonal elements of the form $\chi_{ab,ba}$ show interband transitions from band $b$ to band $a$. Intensity corresponds to the negative of the imaginary part. **b**, Density-density elements of the susceptibility matrix at high frequency. The color scale represents the negative of the imaginary part. **c**, Density-density elements of the susceptibility matrix at low frequency. The color scale represents the negative of the imaginary part.