## [Peer Review File · Nature]

Manuscript Title: Pines' demon observed as a 3D acoustic plasmon in Sr₂RuO₄

Reviewer Comments & Author Rebuttals

Reviewer Reports on the Initial Version:

Referees' comments:

Referee #1 (Remarks to the Author):

Husain et al. performed momentum-resolved electron energy-loss spectroscopy (M-EELS) measurements on Sr₂RuO₄ crystals. They found a low energy electronic mode in the 0~80 meV range with (mostly) linear frequency-momentum dispersion. They identified this mode as Pines' "Demon" predicted 66 years ago, which is an almost charge neutral mode, an out of phase oscillation of local charge densities from two electron pockets. There are two evidences. Firstly, the speed of this mode is too high to be explained by any phonons, while it roughly matches the Demon mode from RPA calculations. Secondly, its spectral weight in the M-EELS spectra scales as a high enough power of momentum, like that of a charge neutral mode. The potential experimental observation of Pines' Demon is important. The evidences provided in this manuscript look promising. To make the conclusion more convincing, I hope the following questions/suggestions could be addressed:

1, Why is the Demon mode from the β - γ pair of bands but not any other pair of bands. For example, the α - γ pair of bands have very different fermi velocities, allowing a potential Demon mode whose velocity lies in between them. Are there qualitative arguments for this choice?

2, The statement on lines 177~179 doesn't look quite rigorous. As stated in Pines' 1956 paper, there is also some constraint on the relative sizes (or plasma frequencies) of the two fermi pockets. Otherwise, the predicted Demon mode may not have a velocity in between the fermi velocities of the two fermi pockets. This would mean that the frequency-momentum of the "Demon" mode lies outside of the range of validity of the approximations made there, meaning that the mode does not exist.

3, Following 2, the fermi velocities of the γ and β bands (from, e.g., Fig. S12) are not super different. Although the charge susceptibility from RPA says there is indeed a dispersion peak lying between these two velocities, the requirement in Pines' 1956 paper (Eq. 25 there in) is only very weakly satisfied. Does this mean the Demon mode measured here is not quite well defined but almost overdamped?

4, Line 89: "Fig. 1" should read "Fig. 2".

5, Line 116: The authors don't really mean the f-sum rule is violated. To avoid confusion, it is better to clarify that this mode seems to violate the sum rule if one only looks at the low frequency range, excluding the plasmon.

6, Line 141: Other than temperature dependent transport data, is there a convincing argument why RPA should be more quantitative in the low energy regime? This contradicts intuition because the effect of electron-electron scattering rate is usually more important in the low frequency range of response functions since the former is relatively large compared to the frequency of interest. For example, hydrodynamical behavior is important in low frequency response functions while not important in the high frequency range.

7, Experimentally, how does the spectra weight of the Demon mode compare to the plasmon? The ration is probably very small, but does it match the prediction of RPA?

Referee #2 (Remarks to the Author):

Husain et al. report on the observation of a collective excitation, known theoretically for more than 60 years, but here claimed to have been seen experimentally for the first time.

The excitation is the "Pines' demon", a special class of bulk plasmon involving different energy bands, and where the electron of one band is oscillating in antiphase with the others.

The authors show some experimental data in general agreement with a RPA model they develop.

In particular, the linear dispersion of the mode, as well as the dependence of the intensity on the momentum are convincing. In fact, the physical explanation of this q dependence in terms of absence of screening is relatively independent on the model and therefore quite robust.

Nevertheless, I can see few issues.

The authors justify that the excitation they see is not a surface plasmon on page 6. However, the explanation is weird: a surface plasmon disperses with energy, therefore one cannot attribute a single value to it ("Richie frequency"). The fact that they don't see a feature at 1.4 eV is therefore not a justification. The authors should explain this. They should also explain which q dependence of the intensity they expect for a surface plasmon, that would be different (and experimentally significantly different) from the Demon behaviour.

It is also not clear why the authors claim to have a bulk feature in their spectrum, when on page 6 the technique is clearly described as a surface one?

Of less importance for the science but maybe worth to enhance for the reader's experience. Could the authors elaborate on why the mode has never been seen before? It is an experimental reason, for sure. Therefore, does this come from the availability of a new material/new preparation, or some sort of technical advance that made it possible? This kind of spectral and q resolution seem to be quite feasible even with a electron *microscope*, so I guess only the spectral range was challenging wr to the state of the art apparatus?

Finally, it seems to me that the Demon is a Goldstone mode, at the opposite of the regular plasmon that can't be a Goldstone mode due to the long-range of the Coulomb interaction. If so, it would be worth mentioning, and in the contrary, explain why it is not.

Author Rebuttals to Initial Comments:

Referee #1 (Remarks to the Author):

Husain et al. performed momentum-resolved electron energy-loss spectroscopy (M-EELS) measurements on Sr₂RuO₄ crystals. They found a low energy electronic mode in the 0~80 meV range with (mostly) linear frequency-momentum dispersion. They identified this mode as Pines' "Demon" predicted 66 years ago, which is an almost charge neutral mode, an out of phase oscillation of local charge densities from two electron pockets. There are two evidences. Firstly, the speed of this mode is too high to be explained by any phonons, while it roughly matches the Demon mode from RPA calculations. Secondly, its spectral weight in the M-EELS spectra scales as a high enough power of momentum, like that of a charge neutral mode. The potential experimental observation of Pines' Demon is important. The evidences provided in this manuscript look promising. To make the conclusion more convincing, I hope the following questions/suggestions could be addressed:

1, Why is the Demon mode from the β - γ pair of bands but not any other pair of bands. For example, the α - γ pair of bands have very different fermi velocities, allowing a potential Demon mode whose velocity lies in between them. Are there qualitative arguments for this choice?

Our Reply:

This is an excellent question. While we do not see it in our experiments, our RPA calculations suggest that there should be a second demon that arises (mainly) from interaction between the α and γ bands. Fig. 1(a) (below) shows line plots of $-\text{Im}[\chi(q,\omega)]$ calculated from RPA for a selection of momenta in the range $0 < q < 0.1$ r.l.u. The large peak corresponds to the β - γ demon that is the subject of our manuscript. In addition, a second mode is present at around 20 meV that forms a shoulder on the main peak:

These modes may be identified by doing a band decomposition of the susceptibility, shown in panels (b) and (c). While the primary mode is composed almost entirely of β and γ electrons (see panel (b)), the 20 meV mode contains significant weight from the α band (see panel (c)), suggesting it is a second demon with α - γ character.

There are three reasons we did not see this α - γ demon in our experiment. First, it contains very little spectral weight because of the small Fermi surface volume of the α band. Second, this mode is at much lower energy, so is harder to resolve from our elastic line. Finally, the α - γ mode is more highly Landau damped, since it resides in a region of (q, ω) in which the particle-hole continuum is stronger.

The existence of a second demon makes sense from the standpoint of mode counting. A metal with one Fermi surface has one longitudinal mode, a plasmon. Sr_2RuO_4 has three Fermi surfaces, so it seems reasonable that it should have three such modes--a conventional plasmon and two demons.

Changes Made:

We added a few sentences to the discussion of Fig. S11 in the supplement identifying and discussing the α - γ mode predicted by RPA.

2, The statement on lines 177~179 doesn't look quite rigorous. As stated in Pines' 1956 paper, there is also some constraint on the relative sizes (or plasma frequencies) of the two Fermi pockets. Otherwise, the predicted Demon mode may not have a velocity in between the Fermi velocities of the two Fermi pockets. This would mean that the frequency-momentum of the "Demon" mode lies outside of the range of validity of the approximations made there, meaning that the mode does not exist.

Our Reply:

Pines' statements about Fermi pocket sizes are not rigid criteria. His 1956 paper is not a definitive analysis of the conditions under which demons exist. It is a physical argument that demons *should* exist, and a description of the conditions he thought would be most favorable for realizing a demon. He only considered the case of parabolic bands with spherical Fermi surfaces, and was guided by his (correct) intuition that, if one subset of electrons was extremely light, it could screen the other and create a neutral mode.

In a real material like Sr_2RuO_4 , which has three bands and significant anisotropy, Pines' simple arguments do not apply. This is why we had to do a microscopic RPA calculation. In Sr_2RuO_4 , the stability of the demon derives from the quasi-1D nature of the β band. As shown in Fig. S10, the particle-hole continuum contains an "eye-shaped" window in which the Lindhard function, $\text{Im}[\chi^0(q, \omega)]$, is small, causing the damping of the demon to be weak. This effect is not considered in Pines' original work, though the demon in Sr_2RuO_4 is gapless and neutral as he predicted.

What is remarkable is that, while the situation Pines considers in his paper is highly simplified, his intuition that demons should exist was correct and applies in real materials with complex band structures.

Changes Made:

We added several sentences to the conclusion as well as to the supplement (including the caption of Fig. S10) explaining that the demon is stable because of the quasi-1D nature of the β band, citing the “eye-shaped” quiet spot in $\text{Im}[\chi^0(q,\omega)]$ illustrated in Fig. S10.

3, Following 2, the fermi velocities of the γ and β bands (from, e.g., Fig. S12) are not super different. Although the charge susceptibility from RPA says there is indeed a dispersion peak lying between these two velocities, the requirement in Pines' 1956 paper (Eq. 25 there in) is only very weakly satisfied. Does this mean the Demon mode measured here is not quite well defined but almost overdamped?

Our Reply:

The answer to point is the same as the previous. Pines' statements about the band velocities are not strict criteria, but general guidelines that derive from the ideal case of two parabolic bands. Understanding whether a demon should be stable in Sr_2RuO_4 requires a microscopic, RPA calculation, which is what we did.

The primary, β - γ demon is not overdamped. It is long lived in Sr_2RuO_4 because of the quasi-1D nature of the β band (see above). This is a situation Pines did not consider.

Changes Made:

Having already explained why the demon is stable in response to the previous comment, this point has already been addressed.

4, Line 89: “Fig. 1” should read “Fig. 2”.

We thank the referee for pointing out this misprint, which has been fixed.

5, Line 116: The authors don't really mean the f-sum rule is violated. To avoid confusion, it is better to clarify that this mode seems to violate the sum rule if one only looks at the low frequency range, excluding the plasmon.

Our Reply:

This is a good point. This statement was an attempt to communicate that the demon is a neutral excitation. But we agree that this way of saying it may be confusing.

Changes Made:

We removed all discussion of sum rule violations from the manuscript. We reworded the sentence in the abstract, and removed the discussion about sum rule violation from the main text. The manuscript is now clearer, as well as a bit shorter.

6, Line 141: Other than temperature dependent transport data, is there a convincing argument why RPA should be more quantitative in the low energy regime? This contradicts intuition because the effect of electron-electron scattering rate is usually more important in the low frequency range of response functions since the former is relatively large compared to the frequency of interest. For example, hydrodynamical behavior is important in low frequency response functions while not important in the high frequency range.

Our Reply:

We did not mean to imply that RPA is a good approximation at small ω . What we are saying is that RPA is a poor approximation at large ω , where vertex corrections, excitonic effects, and local field effects are all important. For example, while RPA correctly predicts the existence of a conventional plasmon at ~ 1.4 eV, it overestimates its lifetime by two orders of magnitude (compare Figs. 2(a) and 3(b)).

At low energy, $\omega < 60$ meV, the comparison is more favorable. Sr_2RuO_4 is a good Fermi liquid in this regime, and RPA is often cited as being exact in Fermi liquid metals at finite ω as $q \rightarrow 0$ (i.e., as the ratio $q/\omega \rightarrow 0$) because it exhausts the sum rule in this limit. RPA predicts a long-lived demon mode with a dispersion that is similar to what we see in our experiment.

That said, the Referee is correct that our experiments also examine the regime $\omega < q$, where hydrodynamic effects may be important. We do not expect RPA to be quantitative in this regime, and we do see important discrepancies between the predicted RPA dispersion and our experiment at low frequency, namely the q^2 "foot" (see Fig. 4 in the main manuscript). We had not previously considered that this could be a hydrodynamic effect, but the Referee is correct that this is a possible explanation.

Changes Made:

We removed all statements from the manuscript that concern fundamental theoretical points about the validity of RPA. This is an experimental paper. The RPA calculation is only meant to serve as a semi-quantitative point of comparison for the data. More general questions about when RPA is a valid approximation lie outside the scope of our paper.

In addition, at the end of the manuscript we now conjecture that the unexplained q^2 foot in the dispersion might be a hydrodynamic effect. We thank the Referee for this suggestion.

7, Experimentally, how does the spectral weight of the Demon mode compare to the plasmon? The [ratio] is probably very small, but does it match the prediction of RPA?

Our Reply:

In both the experiment and in RPA, the plasmon makes up $>99\%$ of the total spectral weight in the f-sum rule. In order to make a quantitative comparison, we performed a partial sum rule

integral over the demon and over the high energy plasmon, both for the experimental data and for our RPA calculation. All the integrals were evaluated for momentum transfer, $q = 0.06$ r.l.u. For the case of RPA, we find the ratio of the plasmon weight to that of the demon to be 1.6×10^4 , while for the M-EELS data the ratio was significantly lower, 4×10^2 . In other words, RPA and the M-EELS experiment agree that the plasmon almost completely exhausts the f-sum rule, since it contains orders of magnitude more spectral weight than the demon. However, they disagree on the number of orders of magnitude.

It's not clear that there is any significance to this quantitative difference. The 3D response function we compute in RPA is not exactly the same as the semi-infinite response function measured with M-EELS [see S. Vig, SciPost Phys. **3**, 026 (2017)]. Also, RPA predictions are poor for the high-energy plasmon, for the reasons discussed above. Quantitative comparisons of this sort are probably pushing the analysis beyond its range of validity.

Changes Made:

None -- The topic of the high energy plasmon and how it relates quantitatively to the demon lies outside the range of validity of RPA and the scope of our paper. So we decided not to add a discussion of this topic to the manuscript.

Referee #2 (Remarks to the Author):

Husain et al. report on the observation of a collective excitation, known theoretically for more than 60 years, but here claimed to have been seen experimentally for the first time. The excitation is the "Pines' demon", a special class of bulk plasmon involving different energy bands, and where the electron of one band is oscillating in antiphase with the others. The authors show some experimental data in general agreement with a RPA model they develop. In particular, the linear dispersion of the mode, as well as the dependence of the intensity on the momentum are convincing. In fact, the physical explanation of this q dependence in terms of absence of screening is relatively independent on the model and therefore quite robust.

Nevertheless, I can see few issues.

The authors justify that the excitation they see is not a surface plasmon on page 6. However, the explanation is weird: a surface plasmon disperses with energy, therefore one cannot attribute a single value to it ("Richie frequency"). The fact that they don't see a feature at 1.4 eV is therefore not a justification. The authors should explain this. They should also explain which q dependence of the intensity they expect for a surface plasmon, that would be different (and experimentally significantly different) from the Demon behaviour.

Our Reply:

Thanks very much to the reviewer for raising this issue. We agree our manuscript should be clearer on the topic of surface plasmons. We give here a quick summary of the subject. For an in-depth review, see W. Plummer, Nucl. Instrum. Meth. B 96, 448 (1995).

A surface plasmon is a collective mode of the boundary of a conductor. If one does a microscopic RPA theory of a surface plasmon, using an instantaneous Coulomb interaction $V = e^2/r$, one finds that a surface plasmon is gapped and has a weak dispersion [see P. J. Feibelman, PRB **9**, 5077 (1974)]. The energy of a surface plasmon as $q \rightarrow 0$ is nonzero and is called the “Ritchie frequency,” ω_R . For an interacting electron gas with a spherical Fermi surface, $\omega_R = \omega_p/\sqrt{2}$, where ω_p is the bulk plasma frequency. In the general case, ω_R is the frequency at which the real part of the dielectric function, $\epsilon(\omega) = -1$.

This picture changes if one assumes the photon velocity is finite. In this case, the Coulomb interaction is no longer instantaneous, and RPA techniques cannot be used since one must explicitly account for retardation of the photon. The simplest approach to this case is the classical electromagnetic analysis introduced by E. N. Economu [see Phys. Rev. **182**, 539 (1969)]. Solving Maxwell’s equations at an interface, this approach yields the following surface plasmon dispersion (red line):

Here, the surface plasmon is now gapless and highly dispersive, because it mixes with the photon in the region of the light cone (green line). In this regime, the excitation is usually called a surface plasmon polariton (SPP). Its velocity in the polariton region is close to c .

Turning to EELS experiments, surface plasmons are often easy to measure. But the dispersing SPP region is rarely observed. The reason is that the light cone is confined to a very small

range of momentum that lies far below the resolution of most EELS instruments [for a notable exception, see P. A. Midgley, *Ultramicroscopy* **76**, 91 (1999)]. For example, in the case of Sr_2RuO_4 , for which $\omega_R = 1.4$ eV, the SPP region should occur at momenta $q < 4 \times 10^{-4}$ r.l.u. This is two orders of magnitude lower than our q resolution of ± 0.01 r.l.u., represented by the scale bar in the above plot. Said another way, for kinematic reasons, the low energy electrons used in our experiment are too slow to be sensitive to the finite velocity of the photon. So, while SPPs are dispersive excitations in principle, in practice they appear as gapped, nondispersive excitations in EELS experiments with slow electrons.

To summarize, the gapless excitation we call a demon cannot possibly be a surface plasmon because its velocity, which is of order v_F , is three orders of magnitude too slow. Furthermore, in low-energy EELS a surface plasmon would appear as a gapped, nondispersive excitation at frequency ω_R for kinematic reasons. The gapless excitation we see therefore cannot be a surface plasmon. [For a study of how surface plasmons appear in our own M-EELS experiments, see Kogar, *PRL* **115**, 257402 (2015)].

Concerning the question of the intensity, a surface plasmon is a conventional charge excitation. Its intensity should follow expectations from the f-sum rule, i.e., $I_0(q) \sim q^{-5}$ as per our Eq. 1. The excitation we see has $I_0(q) \sim q^{-1.8}$, establishing it as a neutral, demon excitation.

Changes Made:

We revised our wording about surface plasmons to get to the core issue, which is the velocity of our excitation is three orders of magnitude too small for it to be a surface plasmon. Here is our exact wording:

However, it cannot be a surface plasmon. While surface plasmons are gapless in the polariton regime, their velocity is close to the speed of light, which is three orders of magnitude faster than the excitation we observe [45].

This simplified wording should now be very clear to the reader.

It is also not clear why the authors claim to have a bulk feature in their spectrum, when on page 6 the technique is clearly described as a surface one?

Our Reply

We should have used more precise wording. M-EELS is very sensitive to bulk excitations. The probe depth of M-EELS is given by the inverse of the in-plane momentum transfer [see E. Evans, D. L. Mills, *PRB* **5**, 4126 (1972)]. In the regime in which the demon is observed, our probe depth is ~ 15 nm. So, while M-EELS experiments require carefully prepared surfaces, the technique can equally detect bulk and surface excitations. It would be more accurate to describe M-EELS as a bulk probe of a semi-infinite system to which it couples through its boundary, rather than as a “surface probe”.

Changes Made:

On page 6, rather than stating that M-EELS measures a “surface response function,” we state that M-EELS measures “the response of a semi-infinite system as probed through its boundary...” This wording is more precise and should be less confusing for the reader.

Of less importance for the science but maybe worth to enhance for the reader's experience. Could the authors elaborate on why the mode has never been seen before? It is an experimental reason, for sure. Therefore, does this come from the availability of a new material/new preparation, or some sort of technical advance that made it possible? This kind of spectral and q resolution seem to be quite feasible even with an electron *microscope*, so I guess only the spectral range was challenging wr to the state of the art apparatus?

Our Reply:

What made it possible to observe the demon was performing meV-resolved EELS measurements in a “parallel-beam” mode that provides high momentum resolution. The Referee is correct that state-of-the-art electron microscopes, such as the Nion UltraSTEM, can now achieve <10 meV energy resolution [see O. L. Krivanek, *Nature* **514**, 209 (2014)], which is sufficient for detecting a demon excitation. However, these instruments are momentum-integrating, since they are configured for high spatial resolution, i.e., for making atomic resolution images. A study of Sr_2RuO_4 with one of these microscopes would integrate over the whole dispersion branch of the demon, wiping this feature out of the spectrum.

That said, there is nothing that prevents an UltraSTEM from being configured to achieve high q resolution. These instruments are fully capable of working in parallel-beam mode with a defocused spot, and would very likely be able to observe the demon in Sr_2RuO_4 in this way.

A STEM running in parallel beam mode would, however, no longer provide atomic level imaging. There is an intrinsic tradeoff between high momentum resolution and high spatial resolution from the uncertainty relation, $\Delta x \Delta k \geq \hbar$. In other words, one cannot measure the propagation of a quantum excitation and simultaneously make nice, real-space pictures. Hopefully our study will inspire others in the STEM field, which is dominated by microscopists, to make compromises in the quality of their images in order to make some new discoveries in quantum phenomena.

Changes Made:

We added a paragraph to the end of the manuscript saying what made this discovery possible is the achievement of high q resolution with EELS and that this would be worth doing in a STEM instrument.

Finally, it seems to me that the Demon is a Goldstone mode, at the opposite of the regular plasmon that can't be a Goldstone mode due to the long-range of the Coulomb interaction. If so, it would be worth mentioning, and in the contrary, explain why it is not.

Our Reply:

We researched this issue, and spoke about it at length with several knowledgeable theorists, and have come to the conclusion that a demon is not a Goldstone mode. A demon is of course gapless, which is one of the key properties of a Goldstone mode. However, a demon does not arise from a broken symmetry. Sr_2RuO_4 is just a homogeneous metal without any spontaneous order. So there is no Goldstone mode to speak of.

Changes Made:

None.

Reviewer Reports on the First Revision:

Referees' comments:

Referee #1 (Remarks to the Author):

In the revised manuscript and the rebuttal letter, most of my questions have been addressed. However, there are two remaining issues that prevent me from recommending it publication:

Firstly, the authors have kept the claim on Line 179 that "Any material with more than one Fermi surface could exhibit a demon..." which is too bold. The long wavelength expressions of the Lindhard dielectric function shows that, if the two fermi velocities v_{f1} and v_{f2} are too close, one may never find a zero of the real part of the dielectric function at any frequency between $v_{f1} \cdot q$ and $v_{f2} \cdot q$. Even if one finds a root for moderately different fermi velocities, the mode may be overdamped if there is no quasi 1D behavior like the beta band in Sr₂RuO₄. If the authors would really like to discuss this, please clarify the condition for the existence of this mode.

Secondly, there is another issue related to the damping rate of the 'demon' which I apologize that I didn't notice during the first round. Fig. 4ab and the discussion on line 147 suggest that the linewidth of the 'demon' is 7.6 meV~46.2 meV, even at 300K. In contrast, Fig. 2 of Ref. 32 says the scattering rate (call it γ_{ee}) in the optical conductivity is $\gamma_{ee} > 150$ meV at 290K and at least as large as the frequency itself at lower temperatures. γ_{ee} is the relaxation rate of the current flow caused by interaction effects. Since the 'demon' involves oscillating current flow of electrons from the two pockets, although out of phase, its damping should be determined by roughly the same γ_{ee} . Therefore, it should be overdamped if one considers the e-e scattering effect beyond RPA.

A possible explanation is similar to the hydrodynamics and electrodynamics of the Dirac fluid in graphene: there are both electrons (from beta and gamma pockets) and holes (from the alpha pocket) in Sr₂RuO₄, and γ_{ee} is the electron-hole scattering rate that conserves the total momentum but not the current. The optical conductivity measures the total current, an out-of-phase oscillation between electrons and holes (meaning electrons and holes flow in opposite directions) which suffers from the electron-hole scattering. In contrast, the observed 'demon' may be a charge neutral hydrodynamic mode, an in-phase oscillation of electrons and holes (meaning electrons and holes flow in the same direction) that is immune to γ_{ee} , thus exhibiting much smaller damping rate.

I hope these two issues could be resolved in the next round.

Referee #2 (Remarks to the Author):

The authors have answered all my questions and the paper is, to my opinion, ready for publication.

Referee #3 (Remarks to the Author):

This paper reports an observation of (presumably) an acoustic plasmon in Sr₂RuO₄. While this is an interesting experiment, there are several aspects that prevent me from recommending this paper for your journal in this or revised form.

First, the presentation is highly misleading. David Pines jocularly called the excitation that is now routinely called acoustic plasmon "demon", a wordplay on an acronym "Dynamic electron motion".

The joke did not really stick, even though Marvin Cohen tried to revive it by putting the word "demon" in the title of their paper on s-d acoustic plasmons (but clarified immediately in the abstract that they meant just usual acoustic plasmons), but that was the first and the last attempt to use this terminology (I doubt Pines himself was expecting this jocular terminology to stick). The dramatic emphasis put on this wording in the current manuscript can only mislead the readers. Acoustic plasmons (AP) were a fashionable and intensely studied subject in the 1970s, splashing over into the 80s. Ruvalds, as well as some Russians (Pashitsky, Geilikman), spent decades and generated dozens of papers advancing superconductivity due to AP, although they were later criticized for neglecting the lattice stability. AP are, of course, exactly analogous to acoustic phonons, in the sense that if there are two different types of carriers, then a uniform shift of the heavy carriers triggers complete screening by the light ones. Just like the acoustic phonons, the AP are, naturally, Goldstone modes, as the 2nd referee pointed out, and it was well understood at that time. In fact, I wanted to look up some papers to refer the authors to, and, lo and behold, I found this discussion on stackexchange (a web site where students reply to each other's questions): <https://physics.stackexchange.com/questions/625137/why-phonons-are-goldstone-modes>. Another interesting hit was this encyclopedia (which I had not been aware of): <https://www.elsevier.com/books/encyclopedia-dictionary-of-condensed-matter-physics/poole-jr/978-0-08-054523-3>, which I found online at <https://books.google.com/books/content?id=CXwrqM2hU0EC&pg=PA12&img=1&zoom=3&hl=en&bul=1&sig=ACfU3U14NLP5xYt98J-uHssHHVzVtbzeQg>, where the issue of AP being Goldstone modes is explicitly discussed.

So, this is not nearly as novel and amazing as the authors pretend. However, it is indeed true that AP have been rather elusive, albeit not true that they have never been reported in equilibrium in the bulk. For instance, there was a paper, which I was able to find, despite my rather vague recollections, <http://dx.doi.org/10.1080/14786435.2012.739290>, which reported AP in Pd. It is worth pointing out that it was for a good reason that Pines originally spoke about s-d AP, and Varma about d-f. While AP are theoretically expected in any multiband metal, they become really acoustic in a meaningful interval of momenta only if the masses are dramatically different. In Pd, where Garrity observes AP, the effective masses in s- and d-bands differ by up to an order in magnitude. In my opinion, the most interesting question is what is so special about Sr₂RuO₄ that AP can be observed even though the masses differ by barely a factor of two? This is exactly what Referee was concerned about, and the response, while probably formally correct, is misleading. Pines's discussion is not as oversimplified as the authors put it. His arguments are valid not just for parabolic bands, but for any 3D material (this is why he put it in terms of the plasma frequencies and not effective masses). The fact that Sr₂RuO₄ breaks out of Pines's (and everybody else's) argument due to its 1D band (which seems to be the case) is highly nontrivial and arguably the most interesting finding in the paper. In the current version it's mentioned in passing as something trivial.

Another point: as discussed in the paper, the observed dispersion is distinctly nonlinear, with flattening out below 0.02 rlu (or likely even below 0.03). Since the "acousticity" of the AP is a fundamental, and not a model-dependent property, it cannot be explained out by "many body phenomena such as hydrodynamic, local field, or excitonic effects", nor can it be related to disorder, since the large-q spectra clearly do not extrapolate to zero. To be precise, it DEFINITELY cannot be related to LF, excitons, and disorder. I am not that sure about hydrodynamics, since I do not understand this well enough, but I suspect that the reason for such a Goldstone mode being so fundamental, it also can only change its velocity and/or damp it, but no shift the linear part to the right (i.e., make it $v^*(q-q_c)$ rather than v^*q). If the authors want to insist on the hydrodynamic origin of such a shift, they should give a more detailed explanation, or to admit that their spectra are in fundamental contradiction with the AP plasmon physics.

Given this question, I am not 100% convinced that what they see is indeed an AP (it may be something more interesting, for all I know...)

A technical remark: What EELS actually measures is $-\text{Im } 1/\epsilon$, which is of course proportional to χ''/q^2 , shown in some figures, but not in the key Fig. 2, which is somewhat misleading.

Author Rebuttals to First Revision:

Referee #1 (Remarks to the Author):

In the revised manuscript and the rebuttal letter, most of my questions have been addressed. However, there are two remaining issues that prevent me from recommending it publication:

Firstly, the authors have kept the claim on Line 179 that “Any material with more than one Fermi surface could exhibit a demon...” which is too bold. The long wavelength expressions of the Lindhard dielectric function shows that, if the two fermi velocities v_{f1} and v_{f2} are too close, one may never find a zero of the real part of the dielectric function at any frequency between $v_{f1} \cdot q$ and $v_{f2} \cdot q$. Even if one finds a root for moderately different fermi velocities, the mode may be overdamped if there is no quasi 1D behavior like the beta band in Sr_2RuO_4 . If the authors would really like to discuss this, please clarify the condition for the existence of this mode.

Our Reply:

This is a fair point. We were attempting to say that materials other than Sr_2RuO_4 may also exhibit demons. But the Referee is correct that, in any given multiband metal, the existence of a demon is not guaranteed.

Changes Made:

To address this point, we added the following paragraph to the closing discussion:

Not every multiband metal is guaranteed to exhibit a demon. Two bands must to be sufficiently different, for example by having different Fermi velocities, to give rise to a distinct pole in the charge response. Further, if Landau damping is strong, the demon may be overdamped and not visible. Nevertheless, the conditions for forming a demon are not unique to Sr_2RuO_4 , and may be present in many materials.

Secondly, there is another issue related to the damping rate of the ‘demon’ which I apologize that I didn’t notice during the first round. Fig. 4ab and the discussion on line 147 suggest that the linewidth of the ‘demon’ is 7.6 meV~46.2 meV, even at 300K. In contrast, Fig. 2 of Ref. 32 says the scattering rate (call it γ_{ee}) in the optical conductivity is $\gamma_{ee} > 150$ meV at 290K and at least as large as the frequency itself at lower temperatures. γ_{ee} is the relaxation rate of the current flow caused by interaction effects. Since the ‘demon’ involves oscillating current flow of electrons from the two pockets, although out of phase, its damping should be determined by roughly the same γ_{ee} . Therefore, it should be overdamped if one considers the e-e scattering effect beyond RPA.

A possible explanation is similar to the hydrodynamics and electrodynamics of the Dirac fluid in graphene: there are both electrons (from beta and gamma pockets) and holes (from the alpha pocket) in Sr_2RuO_4 , and γ_{ee} is the electron-hole scattering rate that conserves the total momentum but not the current. The optical conductivity measures the total current, an out-of-phase oscillation between electrons and holes (meaning electrons and holes flow in opposite directions) which suffers from the electron-hole scattering. In contrast, the observed ‘demon’ may be a charge neutral hydrodynamic mode, an in-phase oscillation of electrons and holes (meaning electrons and holes flow in the same direction) that is immune to γ_{ee} , thus exhibiting much smaller damping rate. I hope these two issues could be resolved in the next round.

Our Reply

This is an excellent point, which we discussed briefly in our previous manuscript but should have emphasized more. The damping of the β - γ demon in Sr_2RuO_4 is anomalously weak because of the quasi-1D character of the β band.

As the Referee suggests, dissipation in a Fermi liquid is caused by four-body scattering near the Fermi surface, $E_1 + E_2 \rightarrow E_3 + E_4$, which results in a scattering rate $\tau^{-1} \propto (\hbar\omega)^2 + (p\pi k_B T)^2$ (e.g., Ashcroft & Mermin, Ch. 17). As shown in D. Stricker, PRL **113**, 087404 (2014) (Ref. 32), the scattering rate in Sr_2RuO_4 exhibits precisely this form, with τ^{-1} at low energy ranging from ~ 10 meV to 150 meV between $T=9\text{K}$ and 290K . This behavior is evidence that Sr_2RuO_4 is a good Fermi liquid at energy scales < 150 meV.

For a collective mode, the analogous damping process is decay into electron-hole pairs, $E_{\text{mode}} \rightarrow E_3 + E_4$, i.e., Landau damping. For the case of the β - γ demon in Sr_2RuO_4 , Landau damping is greatly reduced because of the quasi-1D character of the β band. We illustrate this in Fig. S10 of our Supplementary Information, reproduced in Figure R1 below.

Fig. R1 shows the particle-hole continuum (i.e., $\text{Im}\chi^0(q, \omega)$) in Sr_2RuO_4 computed in RPA, plotted against momentum and energy (see Section II of the Supplementary Information). Because of the quasi-1D character of the β band, the continuum exhibits an eye-shaped “quiet spot” in which the two-particle density of states is reduced.

Perhaps by accident, the dispersion of the demon in Sr_2RuO_4 lies within this quiet spot. The number of accessible decay channels is therefore greatly reduced compared to a typical electron near the Fermi surface. The damping parameter of the demon therefore ranges only from 8 to 46 meV (depending upon q), compared to $\tau^{-1} \sim 150$ meV measured in optics.

Figure R1: Lindhard continuum in Sr_2RuO_4 computed in RPA, showing the eye-shaped quiet spot in which Landau damping of the demon is reduced.

In this sense, there was some luck involved in our observation of the demon in Sr_2RuO_4 . If the β band dispersion were different, the demon might have been overdamped.

Changes Made:

We have clarified this point by adding a new paragraph on how the quasi-1D nature of the β band leads to reduced damping of the demon:

In the case of Sr_2RuO_4 , the demon exists, in part, because of the quasi-1D nature of the β band, which creates an “eye-shaped” region in (q, ω) space in which the two-particle

density of states is reduced (Supplementary Information, Fig. S10). The dispersion curve of the demon, for reasons that may be accidental, lies in this region, causing the demon to be weakly damped compared to other excitations near the Fermi surface [32].

Referee #2 (Remarks to the Author):

The authors have answered all my questions and the paper is, to my opinion, ready for publication.

Our Reply:

We thank this Referee for their effort and thoughtful comments, which have greatly improved our manuscript.

Referee #3 (Remarks to the Author):

This paper reports an observation of (presumably) an acoustic plasmon in Sr₂RuO₄. While this is an interesting experiment, there are several aspects that prevent me from recommending this paper for your journal in this or revised form.

First, the presentation is highly misleading. David Pines jocularly called the excitation that is now routinely called acoustic plasmon “demon”, a wordplay on an acronym “Dynamic electron motion”. The joke did not really stick, even though Marvin Cohen tried to revive it by putting the word “demon” in the title of their paper on s-d acoustic plasmons (but clarified immediately in the abstract that they meant just usual acoustic plasmons), but that was the first and the last attempt to use this terminology (I doubt Pines himself was expecting this jocular terminology to stick). The dramatic emphasis put on this wording in the current manuscript can only mislead the readers.

Our Reply:

The reason we do not use the term “acoustic plasmon” is that it would be misleading. We fully appreciate the Referee’s point about the prevailing terminology used in the 1970’s. However, in 2022—half a century later—the term “acoustic plasmon” refers almost exclusively to plasmons in 2D materials, e.g., single and twisted bilayer graphene, exfoliated transition metal dichalcogenides, surface states in topological insulators, interface 2DEGs in oxides such as LaTiO₃-SrTiO₃, quantum Hall systems, etc. As the Referee is surely aware, even conventional, single-component plasmons are acoustic in two dimensions, for reasons that have little to do with the gapless mode we observed in Sr₂RuO₄. The field of 2D materials hardly existed in the 1970’s, but today it is a major industry.

It was therefore necessary for us to use different terminology that would properly distinguish the excitation in Sr₂RuO₄ from conventional plasmons in 2D materials. The simplest action was to adopt to the terminology Pines designated in his original article [D. Pines, Can. J. Phys. **34**, 1379 (1956)].

The Referee suggests that Pines’ use of the word “demon” was a joke. Pines worked in our department for nearly 60 years, and the prevailing view here is that he was serious about this name. These are rumors. Having passed away in 2018, Pines is no longer here to clarify his intentions. We therefore take him at his word, and use the terminology he wrote down in his paper.

Changes made:

None.

Acoustic plasmons (AP) were a fashionable and intensely studied subject in the 1970s, splashing over into the 80s. Ruvalds, as well as some Russians (Pashitsky, Geilikman), spent decades and generated dozens of papers advancing superconductivity due to AP, although they were later criticized for neglecting the lattice stability.

AP are, of course, exactly analogous to acoustic phonons, in the sense that if there are two different types of carriers, then a uniform shift of the heavy carriers triggers complete screening by the light ones. Just like the acoustic phonons, the AP are, naturally, Goldstone modes, as the 2nd referee pointed out, and it was well understood at that time. In fact, I wanted to look up

some papers to refer the authors to, and, lo and behold, I found this discussion on stackexchange (a web site where students reply to each other's questions):

<https://physics.stackexchange.com/questions/625137/why-phonons-are-goldstone-modes>.

Another interesting hit was this encyclopedia (which I had not been aware of):

<https://www.elsevier.com/books/encyclopedia-of-condensed-matter-physics/poole-jr/978-0-08-054523-3>, which I found online at

<https://books.google.com/books/content?id=CXwrqM2hU0EC&pg=PA12&img=1&zoom=3&hl=en&bul=1&sig=ACfU3U14NLP5xYt98J-uHssHHVzVtbzeQg>,

where the issue of AP being Goldstone modes is explicitly discussed.

Our Reply:

We have carefully reviewed the web sites the Referee linked, as well as many of the published works they cite. This exercise was informative and we appreciate the points made by the Referee. We wish to point out, however, that the question of whether a demon is or is not a Goldstone mode lies outside the scope of our paper.

Our manuscript presents an experiment. We observed a new electronic collective mode in Sr_2RuO_4 that was never seen before. This mode is acoustic and neutral, which are the defining properties of a demon. We have ruled out other plausible interpretations—acoustic phonons, surface plasmons, artifacts due to surface reconstructions, etc. These are experimental facts, and demonstrate that the excitation we see is a demon in the sense that Pines originally defined it.

Our manuscript makes no reference to Goldstone's theorem, which does not arise in the presentation or analysis of our data.

The Referee's comment illustrates the exciting debates, about Goldstone's theorem and many other topics, that will take place if our paper is published.

Changes Made:

None

So, this is not nearly as novel and amazing as the authors pretend.

Our Reply

This one-sentence comment deserves its own response. The Referee is asserting that, because demons are theoretically expected to exist (by Goldstone's theorem, presumably), observing one experimentally for the first time is neither novel nor amazing.

We wish to point that part of the purpose of experiment is to test and validate theories. This is necessary because theories can be wrong or incomplete. A demon has never been observed before experimentally. Seeing one is an important validation of theories of condensed matter and opens the door to new kinds of measurements on multiband metals. Our study is therefore highly novel and important.

Changes Made:

None.

However, it is indeed true that AP have been rather elusive, albeit not true that they have never been reported in equilibrium in the bulk. For instance, there was a paper, which I was able to find, despite my rather vague recollections, <http://dx.doi.org/10.1080/14786435.2012.739290>, which reported AP in Pd.

Our Reply:

This paper by Patrick Garrity on Pd has been discredited. This is why it has zero citations, despite claiming the first experimental observation of a demon / acoustic plasmon.

The Garrity paper, which has only one author, is based on a misunderstanding of the formula for Johnson noise in a conductor. The energies shown in their Fig. 2, which allegedly represent the dispersion of the demon, were incorrectly taken from the amplitude, rather than the frequency, of the voltage fluctuations shown in their Fig. 1. The voltage in a resistance measurement can be set to any value by adjusting the bias current, and has no physical relationship to the energy of the excitation. The paper is riddled with errors, incorrectly labeled figures, and inconsistencies.

Furthermore, the transformer preamp used in this experiment has a bandwidth of 40 kHz, which is *ten orders of magnitude* too slow to observe the excitations claimed in Fig. 2. So the phenomena reported in this manuscript could never have been measured with their setup, even in principle.¹

Note that, after publishing this paper, its sole author left science and never published a paper again. This work should not be cited in an evaluation of our manuscript.

Changes made:

None.

It is worth pointing out that it was for a good reason that Pines originally spoke about s-d AP, and Varma about d-f. While AP are theoretically expected in any multiband metal, they become really acoustic in a meaningful interval of momenta only if the masses are dramatically different. In Pd, where Garrity observes AP, the effective masses in s- and d-bands differ by up to an order in magnitude. In my opinion, the most interesting question is what is so special about Sr₂RuO₄ that AP can be observed even though the masses differ by barely a factor of two? This is exactly what Referee was concerned about, and the response, while probably formally correct, is misleading.

Our Reply:

This is an important point that was also raised by Referee #1 (see above). We mentioned it briefly in our previous manuscript, but we should have emphasized it more. What is special about Sr₂RuO₄ is the quasi-1D character of the β band, which causes the damping of the demon to be anomalously weak. Our response to Referee #1 (above) explains this in detail. Here is a quick summary:

The primary decay channel for a collective mode in a Fermi liquid is Landau damping, i.e., decay into particle-hole pairs. The strength of Landau damping is quantified by the Lindhard function, $\text{Im} \chi^0(q, \omega)$, which parameterizes the two-particle density of states as a function of momentum and energy. The Lindhard function for Sr₂RuO₄, calculated in RPA, is shown in Figure S10 (see Section II of the Supplementary Information), and is also reproduced above in the reply to Referee

¹ Thanks to Prof. Michael Weissman, an expert in resistance fluctuations in metals, for input on this point.

#1. Because of the quasi-1D character of the β band, the continuum exhibits an eye-shaped “quiet spot” in which the two-particle density of states is reduced. The dispersion curve of the demon, perhaps coincidentally, resides in this quiet spot, causing its Landau damping to be highly suppressed. This causes the demon to be well-defined, even though the velocities of the β and γ bands are not exceedingly different.

Changes Made:

We added a new paragraph to the manuscript explaining the Lindhard continuum and the anomalously low damping of the demon (see reply to Referee #1 above). We were able to add this paragraph without exceeding the length limit of 2700 words. We thank both referees for this helpful input, which has made our manuscript clearer and more precise.

Pines’s discussion is not as oversimplified as the authors put it. His arguments are valid not just for parabolic bands, but for any 3D material (this is why he put it in terms of the plasma frequencies and not effective masses). The fact that Sr2RuO4 breaks out of Pines’s (and everybody else’s) argument due to its 1D band (which seems to be the case) is highly nontrivial and arguably the most interesting finding in the paper. In the current version it’s mentioned in passing as something trivial.

Our Reply:

We appreciate this point, and have followed the Referee’s advice by expanding our explanation of Landau damping and the quasi-1D β band, as described above.

Another point: as discussed in the paper, the observed dispersion is distinctly nonlinear, with flattening out below 0.02 rlu (or likely even below 0.03). Since the “acousticity” of the AP is a fundamental, and not a model-dependent property, it cannot be explained out by “many body phenomena such as hydrodynamic, local field, or excitonic effects”, nor can it be related to disorder, since the large- q spectra clearly do not extrapolate to zero. To be precise, it DEFINITELY cannot be related to LF, excitons, and disorder. I am not that sure about hydrodynamics, since I do not understand this well enough, but I suspect that the reason for such a Goldstone mode being so fundamental, it also can only change its velocity and/or damp it, but no shift the linear part to the right (i.e., make it $v^*(q-q_c)$ rather than v^*q). If the authors want to insist on the hydrodynamic origin of such a shift, they should give a more detailed explanation, or to admit that their spectra are in fundamental contradiction with the AP plasmon physics.

Given this question, I am not 100% convinced that what they see is indeed an AP (it may be something more interesting, for all I know...)

Our Reply:

This is not a fair criticism. To summarize the Referee’s point, the Referee asserts that the mode we see is not actually acoustic, because its dispersion is nonlinear. Further, no beyond-RPA effects could ever shift the high-energy part of the dispersion curve, therefore the mode we see is not a demon. The first statement simply is not true. The second ignores all the experimental evidence we present in our paper, and is so sweeping it cannot possibly be justified.

Concerning the first point, it is perfectly allowable for an acoustic mode to exhibit quadratic dispersion at small q . A textbook example is the magnon, the Goldstone mode of a ferromagnet,

whose dispersion (in 1D) is $\omega(q) = 4JS(1 - \cos qa)$. This curve is gapless, quadratic at small q , and crosses over at larger q to something more linear that extrapolates to a nonzero q_0 at zero energy. Nonlinear dispersions are perfectly allowable for a Goldstone mode or other gapless acoustic mode.

Concerning the second point, the Random Phase Approximation (RPA), used in our paper and in Pines' original work, is an *approximation*. It neglects many effects: local field factors, excitonic effects, vertex corrections, self-energy corrections, etc. All of these corrections are frequency- and momentum-dependent, and any could, in principle, displace or distort the dispersion curve of the demon away from a linear form. Excitonic effects, for example, almost always shift excitations to lower energy, as we observed. There is no way the Referee could possibly predict what all these corrections might do without calculating them explicitly. This is particularly true for Sr_2RuO_4 , which has three different valence bands that are nested and highly anisotropic.

Let's briefly summarize the facts established by our experiment:

1. We have observed a new acoustic collective mode in Sr_2RuO_4 that was not seen before.
2. This mode is electronic.
3. This mode is neutral.
4. This mode is not a phonon, a surface plasmon, due to surface reconstructions, or other experimental artifacts.

Taken together, these experimental facts establish the excitation as a demon in the sense that Pines originally defined it. No additional theory is necessary.

Nevertheless, as a reality check, we also performed an RPA calculation and found that Sr_2RuO_4 is *expected* to exhibit a demon. The difference between the RPA and the experimental dispersion curves is less than 12 meV over the entire momentum range. Considering the simplicity of the random phase approximation, the agreement is remarkable, placing our conclusion beyond any reasonable doubt.

What the Referee is arguing is that, because the agreement between our experiment and RPA is not *exact*, the experiment is wrong, and we should ignore the facts (1-4) above. This is not scientific reasoning. When theory and experiment do not agree exactly, it is the theory that needs to be improved.

More broadly, this is an experimental paper. Explaining which higher order corrections are needed to achieve consistency with RPA is not within the scope of our study. This is a job for theorists, and will only occur when our paper is published.

The Referee's comments are another example of the exciting debates our experiment will inspire once our paper is published.

Changes Made:

We updated the closing paragraph about the nonlinear dispersion to also mention vertex corrections, which were not mentioned in the previous version.

A technical remark: What EELS actually measures is $-\text{Im } 1/\epsilon$, which is of course proportional to χ''/q^2 , shown in some figures, but not in the key Fig. 2, which is somewhat misleading.

Our Reply:

We thank the Referee for this helpful observation. The ratio $\chi''(q,\omega)/q^2$ is plotted in both Fig. 2(a) and Fig. 2(b), as indicated in the plot labels. However, we neglected to state this in the figure caption. This was an oversight.

Changes Made:

We have revised the caption of Fig. 2 to explain what quantities are plotted in each of the panels. This revision has improved the overall readability of our manuscript.

Reviewer Reports on the Second Revision:

Referees' comments:

Referee #1 (Remarks to the Author):

The revised manuscript and the rebuttal letter have resolved the first issue but not the damping rate issue raised in my previous report. The damping issue brings a major flaw to the interpretation of the experiment in terms of the Pine's Demon. Therefore, I cannot recommend the publication of the current manuscript.

Since the authors misunderstood the damping issue I was talking about, I try to put it in a clearer way in the following.

At the level of RPA, I have no problem with the authors' explanation in terms of the 'quiet spot'. However, my point is that, γ_{ee} (the electron-electron scattering rate) should bring large dissipation to the Pine's Demon at the level beyond RPA. Specifically, γ_{ee} should appear in self energy and vertex corrections for the response function.

To make an analogy in terms of the optical conductivity, at the RPA level in a single band system, there is no dissipation at nonzero frequency and zero momentum, since this frequency-momentum is outside of the region for Landau damping, similar to what the authors argued. However, there will be dissipation if one considers self energy and vertex corrections. The leading process is decay of the oscillating current by excitation of two electron-hole pairs. This is what the γ_{ee} (scattering rate) means in the optical conductivity measured in Ref. 32.

The γ_{ee} in the optical conductivity is interpreted as a damping rate for the current flow. Since the demon involves flow of electrons (although a counterflow of two pockets), it should suffer a damping rate at the same order. Ref. 32 shows that this damping channel beyond RPA is likely huge, making the demon overdamped. Therefore, if the authors would like to claim the observation of the demon, they must explain why γ_{ee} doesn't overdamp this mode at the level beyond RPA.

Referee #3 (Remarks to the Author):

The paper has been considerably improved. While I still disagree with the authors discarding the "foot" issue as irrelevant, I appreciate that now they relegate it to vortex corrections (which may be causing this "foot"), rather than their previous wording, which I considered incorrect. I also appreciate now that the experimental paper on Pd had some history I was not aware of.

One important part is their insistence of using the term "Pines' demon". I cannot argue with the claim that they had this from the horse's mouth, but the fact remains that Pines never used this term in his subsequent publications, and, more importantly, that there is a large community of condensed matter physicists that know the phenomenon as "3D acoustic plasmon" and would not recognize in the disguise without reading the paper. I could sign off on this paper, if they at least add in the title in parentheses ("acoustic plasmon"), like "Observation of Pines' Demon (3D acoustic plasmon) in Sr₂RuO₄" and add in the abstract and/or introduction the argumentation they present in their response, namely, that these excitations, while formally plasmonic and acoustic, have rather different physical nature from the acoustic plasmons in 2D systems and the authors prefer separating them terminologically. On a personal level, I would not like that, but, objectively, the authors have a valid point that in the last decade or two the term AP has been applied nearly

exclusively to 2D systems (as 50 years ago it was applied nearly exclusively to those "Pines' demons"). However, a clarification to this effect, as above, is indispensable.

Author Rebuttals to Second Revision:

Referee #1 (Remarks to the Author):

The revised manuscript and the rebuttal letter have resolved the first issue but not the damping rate issue raised in my previous report. The damping issue brings a major flaw to the interpretation of the experiment in terms of the Pine's Demon. Therefore, I cannot recommend the publication of the current manuscript.

Since the authors misunderstood the damping issue I was talking about, I try to put it in a clearer way in the following.

At the level of RPA, I have no problem with the authors' explanation in terms of the 'quiet spot'. However, my point is that, γ_{ee} (the electron-electron scattering rate) should bring large dissipation to the Pine's Demon at the level beyond RPA. Specifically, γ_{ee} should appear in self energy and vertex corrections for the response function.

To make an analogy in terms of the optical conductivity, at the RPA level in a single band system, there is no dissipation at nonzero frequency and zero momentum, since this frequency-momentum is outside of the region for Landau damping, similar to what the authors argued. However, there will be dissipation if one considers self energy and vertex corrections. The leading process is decay of the oscillating current by excitation of two electron-hole pairs. This is what the γ_{ee} (scattering rate) means in the optical conductivity measured in Ref. 32.

The γ_{ee} in the optical conductivity is interpreted as a damping rate for the current flow. Since the demon involves flow of electrons (although a counterflow of two pockets), it should suffer a damping rate at the same order. Ref. 32 shows that this damping channel beyond RPA is likely huge, making the demon overdamped. Therefore, if the authors would like to claim the observation of the demon, they must explain why γ_{ee} doesn't overdamp this mode at the level beyond RPA.

Our Reply:

We appreciate now that the Referee was referring to beyond-RPA, four-body decay process that give rise to a scattering rate in IR optics. Previously, we interpreted the question to be about Landau damping, but this was a misunderstanding. We thank the Referee for their patience and for clarifying this point.

To summarize the referee's question, the optical scattering rate, γ_{ee} , represents the damping of a current. In Sr_2RuO_4 , γ_{ee} , measured from IR, ranges from 20-150 meV depending upon the temperature. The demon excitation we see in Sr_2RuO_4 corresponds (at large wavelengths) to equal and opposite currents of γ and β electrons. Yet damping of the demon is significantly smaller, with γ ranging from 8-46 meV (Fig. S8).

The fundamental question is this: Why would the damping of two currents, which exactly compensate one another, be smaller than that of a single, uncompensated current? We hope this is a more accurate summary of the Referee's question.

The answer is that the two counter-propagating currents associated with a demon are not independent, and are not free to decay in the same way as a current of charged particles from a single band.

In Pines' original conceptualization of a demon, one bath of electrons, let's call them type A, screens the Coulomb interaction between electrons in a second bath, call them type B. In his original paper [Can. J. Phys. **34**, 1379 (1956)], Pines argued that this system can be thought of as a gas of composite quasiparticles (mixtures of A and B) that are effectively charge neutral.

Pines' demon is a collective mode of these quasiparticles, call them $A\oplus B$. Their neutrality is what makes a demon acoustic: There is no long-ranged Coulomb energy to overcome, so (unlike a conventional plasmon) the energy of a demon vanishes in the long wavelength ($q\rightarrow 0$) limit. In this sense a demon is somewhat like a phonon, though what compensates the charge is electrons in a second band, rather than the nuclei.

These neutral, composite $A\oplus B$ particles couple weakly to their environment. A neutral particle has no net charge, so its scattering from other excitations in the system, e.g., other electron-hole pairs, is much weaker than that of a charged particle.

At long wavelengths, a demon represents a current of neutral $A\oplus B$ particles, involving zero net displacement of charge. The Referee correctly points out that this neutral current can be thought of as counter-propagating currents from the two different bands, i.e., A and B electrons flowing in opposite directions.

However, a demon is not the same thing as two incoherent currents. Because a demon arises from the movement of neutral quasiparticles, the two currents must fluctuate coherently—in lock step—in such a way as to maintain neutrality of the constituent quasiparticles.

In other words, a demon is not the same thing as two incoherent currents flowing in opposite directions. For example, it would not be possible to create a demon, in a material that did not host one, by somehow launching particles from different bands in opposite directions. The requisite correlations between the particles, which are required to realize a demon, would be absent.

Returning to the original point, the reason the damping of a demon is smaller than the Drude scattering rate is that it is composed of neutral particles, which do not couple as strongly to their environment as charged particles. The four-body decay processes referenced by the Referee, which are the leading contribution to the scattering rate in the Drude conductivity, are mediated by the Coulomb interaction, which for neutral particles is greatly reduced. One therefore expects the decay constant of a demon to be significantly smaller than the scattering rate for a charged current measured in IR optics. This difference is confirmed by our experiment, and further adds to the strength of our conclusion.

It is worth pointing out that the phonons in Sr_2RuO_4 are also significantly sharper than γ_{ee} . Phonons involve currents of valence electrons, yet the acoustic phonon linewidths in Sr_2RuO_4 range from 4-10 meV [see Braden, PRB **76**, 014505 (2007)]. The reason, analogous to the demon, is that the valence electron current is compensated by a current from the nuclei, with which it is highly correlated.

To summarize, one expects the damping of a demon excitation to be significantly smaller than the scattering rate measured with optics, and our experiments confirm this. Together with the other evidence we have already presented—that this excitation is gapless, neutral, and does not arise from surface reconstructions or other artifacts—the smaller damping of the excitation compared to the Drude scattering rate even further strengthens our conclusion that the excitation we see is a demon, which Pines predicted in 1956 and not seen definitively until now.

Changes Made:

We added the following sentence to the closing paragraph on the damping of the demon:

The neutrality of a demon also causes it to couple weakly to other excitations in the system, further enhancing its lifetime.

Referee #3 (Remarks to the Author):

The paper has been considerably improved. While I still disagree with the authors discarding the "foot" issue as irrelevant, I appreciate that now they relegate it to vortex corrections (which may be causing this "foot"), rather than their previous wording, which I considered incorrect. I also appreciate now that the experimental paper on Pd had some history I was not aware of.

One important part is their insistence of using the term "Pines' demon". I cannot argue with the claim that they had this from the horse's mouth, but the fact remains that Pines never used this term in his subsequent publications, and, more importantly, that there is a large community of condensed matter physicists that know the phenomenon as "3D acoustic plasmon" and would not recognize in the disguise without reading the paper. I could sign off on this paper, if they at least add in the title in parentheses ("acoustic plasmon"), like "Observation of Pines' Demon (3D acoustic plasmon) in Sr_2RuO_4 " and add in the abstract and/or introduction the argumentation they present in their response, namely, that these excitations, while formally plasmonic and acoustic, have rather different physical nature from the acoustic plasmons in 2D systems and the authors prefer separating them terminologically. On a personal level, I would not like that, but, objectively, the authors have a valid point that in the last decade or two the term AP has been applied nearly exclusively to 2D systems (as 50 years ago it was applied nearly exclusively to those "Pines' demons"). However, a clarification to this effect, as above, is indispensable.

Our Reply:

We appreciate the Referee's point that, while terminologies have changed in time, there are still active members of the community who knew this excitation as an "acoustic plasmon." This is a valid point and we agree we should try to make the paper clear to the broadest possible readership.

We would like to point out that the third paragraph of our paper (which begins "Surprisingly, while discussed widely in the theoretical literature...") already contains a discussion summarizing the distinction between a demon and a conventional plasmon in two-dimensions. However, we agree with the Referee that the distinction should be clear from the title, and not require reading the whole paper.

Changes made:

After a great deal of discussion, we decided the best solution is to retitle the paper, *Observation of Pines' demon, a 3D acoustic plasmon, in Sr_2RuO_4* . This uses both terms and should be clear to the broadest possible readership.

Reviewer Reports on the Third Revision:

Referees' comments:

Referee #1 (Remarks to the Author):

In the rebuttal letter and the revised manuscript, the authors tried to explain the abnormally small dissipation rate of the 'demon' by two arguments.

The first argument is that the 'demon', as an almost charge neutral mode, has no charge accumulation in its oscillation such that it does not cause electron-hole excitations. This argument does not make sense to me. The reason is that the damping of a mode (consider either the usual plasmon or an acoustic plasmon), as a harmonic oscillator, does not originate from the damping of the potential energy (e.g., the Coulomb potential of the plasmon or the short range potential of the acoustic plasmon) of the oscillator, but from the friction for the kinetic energy (the collective particle flow experiences friction by losing momentum to particle-hole excitations, given that there is no Galilean invariance). Therefore, being a charge neutral mode does not mean that the damping rate of the 'demon' is smaller than the scattering rate in the optical conductivity.

The second argument by the analogy to the acoustic phonon makes sense. Consider a metal with a single Fermi sea (with non-parabolic band such that there is no Galilean invariance, leading to an e-e scattering broadened Drude optical conductivity), the key feature of an acoustic phonon is that the electrons and ions move together in the same direction. In this mode, the local flow is almost like a boost of the whole system, recovering Galilean invariance, explaining why the e-e scattering doesn't bring friction to the flow. Therefore, to explain the small damping of the 'demon' observed by the authors, it makes more sense to use an electron and a hole pocket moving in the same direction, which one of the pocket analogous to the ions in the acoustic phonon. This is just what I proposed in my second report:

"A possible explanation is similar to the hydrodynamics and electrodynamics of the Dirac fluid in graphene: there are both electrons (from beta and gamma pockets) and holes (from the alpha pocket) in Sr₂RuO₄, and γ_{ee} is the electron-hole scattering rate that conserves the total momentum but not the current. The optical conductivity measures the total current, an out-of-phase oscillation between electrons and holes (meaning electrons and holes flow in opposite directions) which suffers from the electron-hole scattering. In contrast, the observed 'demon' may be a charge neutral hydrodynamic mode, an in-phase oscillation of electrons and holes (meaning electrons and holes flow in the same direction) that is immune to γ_{ee} , thus exhibiting much smaller damping rate."

I hope in the revised manuscript, this issue is treated seriously with reasonable explanations, such that it won't mislead the readers.